# DP-C4: Eliminating Solution Bias in Differentially Private Optimization via Coupled Clipping with Adaptive Thresholds

## Abstract

Differentially private (DP) stochastic optimization algorithms are widely used in privacy-preserving deep learning, where per-sample gradient clipping and noise injection protect sensitive information. However, these operations limit existing DP methods to converge within a constant-radius neighborhood of the first-order stationary point, leading to solution bias and the well-known privacy-utility trade-off. To enhance model utility, we propose a novel framework called DP-C4, which is designed to be error-Consistently-decayed, Coupledly-clipped, solution-Calibrated, and Convergence-guaranteed; this is the first time such a method is proposed. Specifically, it incorporates a carefully designed coupled clipping strategy and adaptive clipping thresholds, ensuring that both clipping bias and noise variance asymptotically vanish, thereby correcting the DP-induced solution bias. Furthermore, we develop a memory-efficient variant that reduces storage complexity without compromising privacy guarantees. We prove that our method converges to the optimum in strongly convex case by properly constructing a Lyapunov function, and to a diminishing neighborhood of the first-order stationary point in nonconvex case. Our theoretical results are supported by numerical experiments.

## 1 Introduction

**Background:** Deep learning have been extensively applied in numerous fields, such as smart homes (Li et al., 2023), transportation (Tahaei et al., 2020), and healthcare (Tang et al., 2019). However, the individual privacy whose information is included in datasets should be protected when the models are actually applied. Therefore, it is important to design privacy-preserving algorithms.

Differential Privacy (DP) (Dwork et al., 2006; Dwork & Roth, 2014) has emerged as the gold standard for privacy-preserving deep learning. It offers provable privacy guarantees that the algorithm learns from sensitive data while limiting the information leaked about any individual sample. To protect the privacy of the training data, numerous differentially private stochastic optimization algorithms have been proposed for deep learning, such as DP stochastic graident descent (DP-SGD) (Abadi et al., 2016). They apply per-sample gradient clipping using a fixed clipping norm and adds Gaussian noise into the aggregated gradient , which have been successfully deployed in both centralized (McMahan et al., 2018b; Bu et al., 2020) and federated (Geyer et al., 2017; Truex et al., 2020) settings.

However, the perturtion introduced by gradient clipping and noise often leads to reduced model accuracy. Therefore, these methods face a trade-off between model utility and privacy (Amin et al., 2019; Zhang et al., 2023a; Xiao et al., 2023). This challenge has attracted considerable attention, leading to the development of several improved variants of DP stochastic optimization algorithms. In particular: (1) adaptive clipping thresholds (Andrew et al., 2021; Phan et al., 2017; Pichapati et al., 2019) are adopted to reduce noise variance; (2) gradient normalization or group-based clipping (Yang et al., 2022; Das et al., 2021; McMahan et al., 2018a) are designed to mitigate clipping bias; and (3) iterative schemes are transferred from advanced non-DP optimizers (Zhu et al., 2024; Murata & Suzuki, 2023; Lee, 2017) to leverage their advantageous properties. Nevertheless, gradient clipping and added noise inevitably alter the original optimization dynamics. Prior work shows

that under settings similar to DP-SGD, regardless of how the clipping threshold or step size is chosen, DP algorithms only *converge with a constant bias term*, i.e., converge to a neighborhood of the first-order stationary point with a constant radius (Chen et al., 2020; Xiao et al., 2023; Song et al., 2013). Recently, the DiceSGD algorithm (Zhang et al., 2023b) integrates an Error Feedback mechanism to eliminate clipping bias at each iteration, enabling convergence *in expectation over the injected noise*. However, it does not account for noise variance, thereby driving the iterates to drift away from the optimum, leaving the solution bias issue. As a result, existing DP algorithms fail to handle both clipping bias and noise variance. This naturally motivates a fundamental but important question:

*Is it possible to design a DP stochastic optimization algorithm that both clipping bias and noise variance asymptotically vanish during iterations, thereby eliminating the issue of solution bias?*

**Our Contributions:** We provide an affirmative answer to the question by proposing an **error-Consistently-vanishing, Coupledly-clipped, solution-Calibrated, and Convergence-guaranteed** (DP-C4) algorithmic framework. This method incorporates a carefully designed coupled clipping strategy and adaptive clipping thresholds, thereby enforcing the clipping bias and noise variance to asymptotically vanish during iterations. To the best of our knowledge, this is the first time such a method is proposed. Furthermore, to mitigate the extra memory cost for determining clipping thresholds, we propose DP-C4$^+$, which ensures a lower memory cost while preserving the calibration property. We prove that our method converges to the optimum in strongly-convex case by properly constructing a Lyapunov function and to a diminishing neighborhood of the first-order stationary point in the nonconvex case. Notably, we derive the upper bound through a case-by-case analysis leveraging the clipping strategy, thereby opening up new avenues for convergence analysis. Specifically, our contributions are as follows:

- **DP-C4 Framework**: We propose DP-C4, the first DP stochastic optimization algorithmic framework that eliminates solution bias by ensuring the joint asymptotic vanishing of noise variance and clipping bias. Furthermore, to reduce memory overhead, we introduce DP-C4$^+$, which matches the memory cost of DP-SGD while preserving the solution calibration benefits of DP-C4.

- **Novel Convergence Analysis**: We establish the convergence guarantees of DP-C4$^{(+)}$. Specifically, this method converges to the optimum by properly constructing Lyapunov functions in strongly-convex case, and to a diminishing neighborhood of the first-order stationary point in nonconvex case. To our best knowledge, this is the first DP algorithm whose convergence can be analyzed via a Lyapunov function, due to its unique solution calibration property.

- **Privacy Guarantee**: We present a privacy budget allocation strategy utilizing the structure of DP-C4$^{(+)}$ to guarantee privacy. Compared to DP-SGD, it can achieve the same level of privacy protection while adding less noise.

- **Empirical Validation**: We conduct extensive experiments showing our method achieves superior privacy-utility trade-offs over existing baselines across various tasks and datasets.

## 2 Preliminaries

### 2.1 Problem Setup and Assumptions

**Problem Setup:** We consider the empirical risk minimization (ERM) problem on a dataset $D$ with $|D| = N$:

$$\min_{x \in \mathbb{R}^d} f(x) := \frac{1}{N} \sum_{i=1}^{N} f_i(x), \tag{1}$$

where $f_i(x)$ denotes the loss associated with the $i$-th data sample. Our goal is to propose a DP stochastic optimization algorithmic framework with Gaussian mechanism for finding its first-order stationary point $x^\star$, i.e., $\nabla f(x^\star) = \frac{1}{N} \sum_{i=1}^{N} \nabla f_i(x^\star) = 0$.

**Definition 1** (($\epsilon, \delta$)-Differential Privacy (Dwork et al., 2006)). *A randomized mechanism $\mathcal{M} : \mathcal{D} \to \mathcal{R}$ is said to satisfy ($\epsilon, \delta$)-DP if for any two neighboring datasets $D, D' \in \mathcal{D}$ differing in at most one*

*data record, and for any measurable subset $\mathcal{S} \subseteq \mathcal{R}$, it holds that*

$$\Pr[\mathcal{M}(D) \in \mathcal{S}] \leq e^\epsilon \Pr[\mathcal{M}(D') \in \mathcal{S}] + \delta. \tag{2}$$

*Here, $\epsilon > 0$ is the privacy budget controlling the strength of privacy protection, and $\delta \in [0,1]$ denotes a negligible probability of failure.*

**Definition 2** (Gaussian Mechanism (Dwork & Roth, 2014)). *Given a function $f : \mathcal{D} \to \mathbb{R}^d$ and dataset $D \in \mathcal{D}$, the Gaussian mechanism adds noise calibrated to the $\ell_2$-sensitivity of $f$:*

$$\mathcal{M}(D) = f(D) + \mathcal{N}(0, \sigma^2 \mathbf{I}_d), \tag{3}$$

*where $\mathcal{N}(0, \sigma^2 \mathbf{I}_d)$ denotes a $d$-dimensional Gaussian distribution with zero mean and covariance $\sigma^2 \mathbf{I}_d$. The noise scale satisfies $\sigma \geq \Delta_f \cdot \frac{\sqrt{2 \log(1.25/\delta)}}{\epsilon}$, with $\Delta_f = \max_{D,D'} \|f(D) - f(D')\|_2$ denoting the $\ell_2$-sensitivity of $f$ between neighboring datasets $D$ and $D'$.*

## 2.2 DP-SGD AND DP-SVRG:

In this subsection, we give a brief review of the DP-SGD and DP-SVRG methods.

**DP-SGD:** DP-SGD (Abadi et al., 2016) is a widely adopted method for solving (1). At $k$-th iteration, it randomly selects a subset $S_k \subseteq D$, clips the $l_2$ norm of each gradient, and then adds noise to protect privary. The iterative scheme with a fixed clipping threshold $C$ is:

$$x^{k+1} = x^k - \frac{\eta}{|S_k|} \sum_{i \in S_k} \left( \text{clip}(\nabla f_i(x^k), C) + \mathcal{N}(0, \sigma^2 C^2 I) \right), \tag{4}$$

where $\eta > 0$ is the step size and $\text{clip}(\nabla f_i(x^k), C) := \nabla f_i(x^k) \min\{1, \frac{C}{\|\nabla f_i(x^k)\|_2}\}$. A more flexible approach is to let the clipping threshold $C_k$ vary. From a noise-reduction perspective, we would like $C_k \to 0$ as $x^k \to x^\star$. However, because stochastic gradients typically have variance, which is nonzero at $x^\star$ ($\|\frac{1}{|S_k|}\sum_{i \in S_k}\nabla f_i(x^\star)\|_2 \neq \|\frac{1}{|D|}\sum_{i \in D}\nabla f_i(x^\star)\|_2 = 0$), $C_k$ can not be set too small during iterations. Therefore, variance reduction techniques for gradient estimation seem to hold promise for enhancing utility in DP algorithms.

| **Algorithm 1** DP-SGD | **Algorithm 2** DP-SVRG |
|---|---|
| 1: Initialize $x^0$ | 1: Initialize $x^0 = w^0$ |
| 2: **for** $k = 0, 1, 2, \ldots$ **do** | 2: **for** $k = 0, 1, 2, \ldots$ **do** |
| 3:  Sample $S_k \subseteq D$ | 3:  Sample $S_k \subseteq D$ |
| 4:  $g_i^k = \text{clip}(\nabla f_i(x^k), C)$ | 4:  $\tilde{g}_i^k(x) = \text{clip}(\nabla f_i(x^k), C) + \mathcal{N}(0, \sigma^2 C^2 I)$ |
| 5:  $\tilde{g}_i^k = g_i^k + \mathcal{N}(0, \sigma^2 C^2 I)$ | 5:  $\tilde{g}_i^k(w) = \text{clip}(\nabla f_i(w^k), C) + \mathcal{N}(0, \sigma^2 C^2 I)$ |
| 6:  $\tilde{g}^k = \frac{1}{|S_k|}\sum_i \tilde{g}_i^k$ | 6:  $\tilde{g}^k = \frac{1}{|S_k|}\sum_{i \in S_k}\tilde{g}_i^k(x) - \frac{1}{|S_k|}\sum_{i \in S_k}\tilde{g}_i^k(w) + \frac{1}{|D|}\sum_{i \in D}\tilde{g}_i^k(w)$ |
| 7:  $x^{k+1} = x^k - \eta\tilde{g}^k$ | 7:  $x^{k+1} = x^k - \eta\tilde{g}^k$ |
| 8: **end for** | 8:  $w^{k+1} = \begin{cases} x^k, & \text{with probability } p \\ w^k, & \text{with probability } 1-p \end{cases}$ |
|  | 9: **end for** |

**DP-SVRG:** The SVRG (Johnson & Zhang, 2013; Kovalev et al., 2020) method is a representative variance reduction technique. This method introduces an additional anchor point $w^k$, which is periodically updated and computed the full gradient. At $k$-th iteration, the gradient estimate is $g_{S_k}^k = \frac{1}{|S_k|}\sum_{i \in S_k}\nabla f_i(x^k) - \frac{1}{|S_k|}\sum_{i \in S_k}\nabla f_i(w^k) + \frac{1}{|D|}\sum_{i \in D}\nabla f_i(w^k)$, which is an unbiased estimate of the full gradient, i.e., $\mathbb{E}[g_{S_k}^k] = \nabla f(x^k)$. Moreover, it satisfies $g_{S_k}^k \xrightarrow{x^k, w^k \to x^\star} \nabla f(x^\star) = 0$. By integrating SVRG into the DP algorithm, DP-SVRG (Lee, 2017) has been proposed (see Alg.2). However, clipping $\nabla f_i(x^k)$ and $\nabla f_i(w^k)$ separately undermines the variance-reduction structure, where the resulting stochastic gradient becomes biased ($\mathbb{E}\tilde{g}^k \neq \nabla f(x^k)$) and no longer converges to zero as $x^k \to x^*$.

In summary, DP algorithms face the following trade-off issue: On the one hand, choosing a large clipping threshold leads to substantial noise injection. On the other hand, a small threshold causes excessive clipping bias of the gradient estimates. In this paper, we focus on designing DP algorithmic framework that handles both clipping bias and noise variance to eliminate solution bias.

## 3 METHOD: DP-C4

In this section, we propose a DP stochastic optimization framework called **DP-C4**, which is **error-Consistently-vanishing, Coupledly-clipped, solution-Calibrated, and Convergence-guaranteed**. This framework ensures the asymptotic vanish of both the noise variance and the clipping bias, thereby eliminating solution bias.

### 3.1 HIGH-LEVEL IDEA

We consider constructing the gradient estimator by aggregating multiple sub-estimators $\{h^{(j)}(x)\}_{j\in[n]}$ that satisfy $\sum_{j\in[n]} \mathbb{E}[h^{(j)}(x)] = \nabla f(x)$. Each sub-estimator is defined by $h^{(j)}(x) := \frac{1}{|S_j|}\sum_{i\in S_j} h_i^{(j)}(x)$, where $S_j \subseteq D$ denotes the sampled dataset and $\{h_i^{(j)}\}_{i\in S_j, j\in[n]}$ denotes per-sample estimators. Furthermore, for DP algorithms, we clip the $l_2$ norm of each component $h_i^{(j)}(x)$, aggregate the clipped components and add noise to form the DP gradient estimator $\tilde{g}^k$. For simplicity, we focus on the case $n = 2$. The iterative scheme is given by:

$$\begin{cases} x^{k+1} = x^k - \eta\tilde{g}^k, \\ \tilde{g}^k = \left[\frac{1}{|S_1|}\sum_{i\in S_1}\text{clip}(h_i^{(1)}(x^k), C_1) + n_1^k\right] + \left[\frac{1}{|S_2|}\sum_{i\in S_2}\text{clip}(h_i^{(2)}(x^k), C_2) + n_2^k\right], \\ n_1^k \sim \mathcal{N}(0, \sigma_1^2 C_1^2 I), \quad n_2^k \sim \mathcal{N}(0, \sigma_2^2 C_2^2 I). \end{cases}$$

Here, $C_j$ is the clipping threshold and $\sigma_j^2$ is the privacy-dependent noise multiplier. Instead of using fixed $C_j$ during iterations, we consider replacing them with an estimator-dependent function $C_j(\{h_i^{(j)}(x^k)\}_{i\in S_j})$, ensuring both clipping bias $B_k^{(j)}$ and noise variance $V_k^{(j)}$ vanish asymptotically as $x^k \to x^\star$:

$$\begin{cases} B_k^{(j)} := \left\|\frac{1}{|S_j|}\sum_{i\in S_j}\text{clip}(h_i^j(x^k), C_j) - \frac{1}{|S_j|}\sum_{i\in S_j}h_i^j(x^k)\right\|^2 \xrightarrow{x^k\to x^\star} 0, \\ V_k^{(j)} := \sigma_j^2 C_j^2(\{h_i^j(x^k)\}) \xrightarrow{x^k\to x^\star} 0, \end{cases}$$

where $||\cdot||$ denotes $l_2$-norm. To guide the design, we first establish an upper bound on clipping bias in Lemma 1 (Proof in Appendix C):

**Lemma 1** (Upper Bound on Clipping Bias). *Let $I_1^k := \{i\in S : \|h_i(x^k)\| < C(\{h_i(x^k)\}_{i\in S})\}$ be the set of unclipped samples, and $I_2^k := \{i\in S : \|h_i(x^k)\| \geq C(\{h_i(x^k)\}_{i\in S})\}$ the clipped ones. Then,*

$$B_k \leq \frac{|I_2^k|}{|S|^2}\sum_{i\in I_2^k}\left[||h_i(x^k)|| - C(\{h_i(x^k)\}_{i\in S})\right]^2.$$

Lemma 1 implies $B_k \to 0$ as both $\|h_i(x^k)\| \to 0$ and $C(\{h_i(x^k)\}_{i\in S}) \to 0$. Therefore, to push the clipping bias $B_k^{(1)}$ to zero as $x^k \to x^\star$, as a natural choice, we set $\{h_i^{(1)}\}_{i\in S_1}$ and $C_1(\{h_i^{(1)}\}_{i\in S_1})$ as:

$$h_i^{(1)}(x^k) := \nabla f_i(x^k) - \nabla f_i(x^\star), \quad C_1(\{h_i^{(1)}(x^k)\}_{i\in S_1}) := C_1\cdot\left\|\frac{1}{|S_1|}\sum_{i\in S_1}\left(\nabla f_i(x^k) - \nabla f_i(x^\star)\right)\right\|,$$

where $C_1$ is a scaling factor. However, since $x^*$ is unknown, we replace $x^*$ with a history iterate $w^k \in \{x^{k-i}\}_{i\in[k]}$:

$$h_i^{(1)}(x^k, w^k) := \nabla f_i(x^k) - \nabla f_i(w^k), \quad C_1(\{h_i^{(1)}(x^k, w^k)\}_{i\in S_1}) := C_1\cdot\left\|\frac{1}{|S_1|}\sum_{i\in S_1}(\nabla f_i(x^k) - \nabla f_i(w^k))\right\|.$$

---

**Algorithm 3** DP-C4

---

1: **Input:** Dataset $\mathcal{D}$, learning rate $\eta$, clipping bounds $C, C_1, C_2$, noise scales $\sigma_1, \sigma_2$, total steps $T$, anchor update probability $p$

2: **Output:** Model parameters $x^T$ satisfying $(\varepsilon, \delta)$-DP

3: **Initialize:** $x^0 = w^0 \in \mathbb{R}^d$

4: **for** $k = 0$ to $T - 1$ **do**

5:      Sample $S \subseteq D$

6:      $C_{1k} \leftarrow \min(C, C_1 \| \frac{1}{|S|} \sum_{i \in S} (\nabla f_i(x^k) - \nabla f_i(w^k)) \|)$          {Coupled threshold}

7:      $C_{2k} \leftarrow \min(C, C_2 \| \nabla f(w^k) \|)$          {Anchor threshold}

8:      $g_1^k \leftarrow \frac{1}{|S|} \sum_{i \in S} \texttt{clip}(\nabla f_i(x^k) - \nabla f_i(w^k), C_{1k})$          {Coupled term}

9:      $g_2^k \leftarrow \frac{1}{|D|} \sum_{i \in D} \texttt{clip}(\nabla f_i(w^k), C_{2k})$          {Anchor term}

10:     $n_1^k \sim \mathcal{N}(0, \sigma_1^2 C_{1k}^2 I), \quad n_2^k \sim \mathcal{N}(0, \sigma_2^2 C_{2k}^2 I)$          {Sample DP noise}

11:     $\tilde{g}^k \leftarrow g_1^k + g_2^k + n_1^k + n_2^k$          {Add noise}

12:     $x^{k+1} \leftarrow x^k - \eta \cdot \tilde{g}^k$          {Update model}

13:     $w^{k+1} \leftarrow \begin{cases} x^k, & \text{with probability } p \\ w^k, & \text{with probability } 1 - p \end{cases}$          {Update anchor (Routine 1)}

14: **end for**

---

When $x^k, w^k \to x^\star$, we have $B_k^{(1)} \to 0$. Meanwhile, $V_k^{(1)} \to 0$ since $C_1(\{h_i^{(1)}\}_{i \in S_1}) \to 0$. For the sub-estimator $h^{(2)} = \frac{1}{|S_2|} \sum_{i \in S_2} h_i^{(2)}$, we choose $S_2 = D$, set $\{h_i^{(2)}\}_{i \in S_2}$ and $C_2(\{h_i^{(2)}\}_{i \in S_2})$ as:

$$h_i^{(2)}(x^k, w^k) := \nabla f_i(w^k), \quad C_2(\{h_i^{(2)}(x^k, w^k)\}_{i \in S_2}) := C_2 \cdot \|\nabla f(w^k)\|,$$

where $C_2$ is a scaling factor. This choice ensures $\mathbb{E}[h^{(1)}(x^k, w^k) + h^{(2)}(x^k, w^k)] = \nabla f(x^k)$, and makes:

$$B_k^{(2)} + V_k^{(2)} \leq (\sigma_2^2 + 1) \cdot C_2^2 \cdot \|\nabla f(w^k)\|^2 \to 0 \quad \text{as } w^k \to x^*.$$

As a result, our proposed gradient estimator and clipping thresholds ensure that all error components (clipping bias, noise variance) asymptotically vanish, which forms the foundation of our DP algorithmic framework.

## 3.2 DP-C4 Algorithm

In this subsection, we formally describe the DP-C4 method in Alg.3. Based on the idea in subsection 3.1, Alg.3 constructs a gradient estimator by aggregating two sub-estimators: a coupledly-clipped gradient difference term (Line 8) and a clipped anchor term (Line 9). Specifically, we initialize with $x^0 = w^0 \in \mathbb{R}^d$. At the $k$-th iteration, we sample a mini-batch $S \subseteq D$ (Line 5). We compute the gradient difference $\nabla f_i(x^k) - \nabla f_i(w^k)$ for $i \in S$, and aggregate them to obtain the clipping threshold $C_{1k}$ (Line 6). Here, an upper bound $C$ is introduced to prevent injecting excessively large noise during the early iterations. Next, we clip each gradient difference and aggregate the clipped values to form the sub-estimator $g_1^k$ (Line 8). Meanwhile, $C_{2k}$ and $g_2^k$ are computed only with probability $p$ since the anchor $w^k$ is updated with probability $p$. Finally, by aggregating $g_1^k$ and $g_2^k$ and adding noise, we obtain the perturbed gradient estimator $\tilde{g}^k$. Moreover, for updating the anchor $w^k$ (Line 13), there are also alternative routines (see the following Routine 2-4):

$$w_{R_2}^{k+1} = \begin{cases} x^k, & k = 1 (mod \lceil 1/p \rceil) \\ w^k, & k \neq 1 (mod \lceil 1/p \rceil) \end{cases}, \quad w_{R_3}^{k+1} = \begin{cases} x^{k+1}, & with\ p \\ w^k, & with\ 1-p \end{cases}, \quad w_{R_4}^{k+1} = \begin{cases} x^{k+1}, & k = 1 (mod \lceil 1/p \rceil) \\ w^k, & k \neq 1 (mod \lceil 1/p \rceil) \end{cases},$$

We emphasize that the DP-C4 method differs fundamentally from the DP-SVRG method (Alg.2). Specifically, DP-C4 focuses on clipping the gradient difference $\nabla f_i(x^k) - \nabla f_i(w^k)$ for each $i \in S_k$, whereas DP-SVRG clips $\nabla f_i(x^k)$ and $\nabla f_i(w^k)$ seperately. Moreover, DP-C4 adaptively determines the clipping threshold. These core distinctions allow DP-C4 to asymptotically vanish both the clipping bias and the noise variance.

## 3.3 Solution-Calibrated Property of DP-C4

Consider the ERM problem (1) and let $x^\star$ denote a solution that satisfies the first-order optimality condition, i.e., $\frac{1}{|D|} \sum_{i \in D} \nabla f_i(x^\star) = \nabla f(x^\star) = 0$. To further demonstrate the desirable properties

of DP-C4, we consider all sources of randomness (i.e., sampling, noise, and anchor-update) and investigate the potential convergence point of Alg.3 from the perspective of fixed-point analysis. Specifically, at a fixed point $(\tilde{x}, \tilde{w})$, both sequences $\{x^k\}_{k \in \mathbb{N}}$ and $\{w^k\}_{k \in \mathbb{N}}$ converge, implying $x^{k+1} = x^k = \tilde{x}, w^{k+1} = w^k = \tilde{w}$. Hence, we substitute it into Alg.3, the fixed point of DP-C4 satisfies the following system:

$$
\begin{cases}
\frac{1}{|S|}\sum_{i \in S}\text{clip}\left(\nabla f_i(\tilde{x}) - \nabla f_i(\tilde{w}), C_{1k}\right) + \frac{1}{|D|}\sum_{i \in D}\text{clip}(\nabla f_i(\tilde{w}), C_{2k}) + \mathbf{n}_1^k + \mathbf{n}_2^k = 0, \\
\tilde{w} = w^{k+1} = \begin{cases} x^k = \tilde{x}, & \text{with probability } p, \\ w^k, & \text{with probability } 1 - p, \end{cases} \\
\mathbf{n}_1^k \sim \mathcal{N}(0, \sigma_1^2 C_{1k}^2 I), \quad \mathbf{n}_2^k \sim \mathcal{N}(0, \sigma_2^2 C_{2k}^2 I), \\
C_{1k} = \min(C, C_1 \| \frac{1}{|S|}\sum_{i \in S}(\nabla f_i(\tilde{x}) - \nabla f_i(\tilde{w}))\|), \\
C_{2k} = \min(C, C_2 \| \frac{1}{|D|}\sum_{i=1}^{|D|} \nabla f_i(\tilde{w})\|).
\end{cases}
\tag{5}
$$

To satisfy this fixed-point system, for the first equation in (5), it must hold that $\mathbf{n}_1^k = \mathbf{0}$ and $\mathbf{n}_2^k = \mathbf{0}$ due to the iteration-wise independence of the noise randomness. This implies:

$$
\left\| \frac{1}{|S|}\sum_{i \in S}(\nabla f_i(\tilde{x}) - \nabla f_i(\tilde{w})) \right\| = 0, \quad \left\| \frac{1}{|D|}\sum_{i=1}^{|D|} \nabla f_i(\tilde{w}) \right\| = 0,
$$

which forces $\tilde{x} = \tilde{w} = x^\star$. Substituting this into (5), all conditions are satisfied. Therefore, it follows that a point is a fixed point of DP-C4 if and only if it is a first-order stationary point of the ERM problem (1), indicating DP-C4 eliminates solution bias.

In contrast, exiting DP algorithms with constant clipping thresholds (e.g., DP-SGD, DP-SVRG) do not admit fixed points, as the fixed-variance noise injected at each iteration continually disrupts equilibrium. For other schemes where clipping thresholds decays to 0, the persistent gradient estimation variance and gradual accumulation of clipping bias, combined with a mismatch between the decay rate of the thresholds and the convergence speed, lead to the fixed point being, with probability 1, not a solution to the original problem. The detailed comparison is provided in Appendix B.

## 3.4 CONVERGENCE ANALYSIS

In this subsection, we analyze the convergence properties of DP-C4 under two settings: (i) $\mu$-strongly convex, and (ii) nonconvex. Our goal is to construct a Lyapunov function in strongly convex case with specific clipping thresholds, and to establish convergence guarantees in non-convex case without restrictions on clipping thresholds. It is worth emphasizing that these proofs are innovative in the following aspects: (1) existing DP algorithms lack solution-calibrated property and thus cannot employ Lyapunov functions for analysis; (2) by exploiting the unique structure of DP-C4, we carefully handle both noise variance and clipping bias, providing a novel perspective for the convergence analysis of DP optimization algorithms. We first present several assumptions:

**Assumption 3.1 (Lower Bounded)** $f(\cdot)$ is bounded from below by a finite constant $f^\star$:

$$
f(x) \geq f^\star > -\infty, \, \forall\, x \in \mathbb{R}^d.
$$

**Assumption 3.2 (L-Smoothness)** $f_i(\cdot)$ is $L$-smooth, i.e., it satisfies:

$$
\|\nabla f_i(x) - \nabla f_i(y)\| \leq L\|x - y\|, \, \forall\, x, y \in \mathbb{R}^d.
$$

**Assumption 3.3 ($\mu$-Strong Convexity)** The loss function $f_i(\cdot)$ is $\mu$-strongly convex:

$$
f_i(y) \geq f_i(x) + \langle \nabla f_i(x), y - x \rangle + (\mu/2)\|x - y\|^2, \, \forall\, x, y \in \mathbb{R}^d.
$$

**Assumption 3.4 (Bounded Variance)** There exists a constant $\tau$, such that:

$$
\|\nabla f_i(x) - \nabla f(x)\| \leq \tau, \quad \forall i \in [N], \forall x \in \mathbb{R}^d.
$$

**Assumption 3.5 (Bounded Gradient).** The gradient of the function is bounded in the sense that there exists a positive constant $G = \sup_{x \in \mathbb{R}^d, i \in [N]} \|\nabla f_i(x)\| < \infty$.

The above assumptions serve as the foundation for analyzing DP algorithms. We now turn to the convergence of DP-C4. To avoid overly intricate discussions, we restrict our setting to $C_{1k} = C_1||\frac{1}{|S|}\sum(\nabla f_i(x^k) - \nabla f_i(w^k))||, C_{2k} = C_2||\nabla f(w^k)||$. Let $\mathbb{E}[\cdot]$ and $\mathbb{E}_k[\cdot] := \mathbb{E}[\cdot|x^k, w^k]$ denote the full expectation and the conditional expectation based on the first $k$ iterations of DP-C4, respectively. Then, we have:

**Theorem 1** (Strongly Convex Case). *Suppose Assumptions 3.1-3.5 hold. For any given $e > 0$ and constant DP noise multipliers $\sigma_1, \sigma_2$, let $\{x^k\}_{k\geq 0}$ and $\{w^k\}_{k\geq 0}$ be generated by Alg.3 with $\eta < \min\left\{\frac{\mu}{3N_1+A}, \frac{1}{2LN_2}\right\}, C_1 > 0, C_2 \geq \frac{\tau}{e} + 1$. When $\min\{||\nabla f(w^k)||, ||x^k - x^\star||\} > e$, define the Lyapunov function as:*

$$\Phi^k := \mathbb{E}||x^k - x^\star||^2 + \frac{2N_1\eta^2}{p}\mathbb{E}||w^k - x^\star||^2 + \frac{2N_2\eta^2}{p}D^k,$$

*where $D^k := \mathbb{E}||\nabla f_i(w^k) - \nabla f_i(x^\star)||^2, N_1 := 8L^2C_1^2(d\sigma_1^2+1), N_2 := 4C_2^2(d\sigma_2^2+1), A := \frac{4G^2}{pe^2\mu^2}(L - C_1\mu)\sqrt{4L^2C_1^2(d\sigma_1^2+1) + \mu^2C_2^2(d\sigma_2^2+1)}$, and $d$ denotes the model size. Then,*

$$\Phi^{k+1} \leq \max\left\{1 - \mu\eta + (3N_1+A)\eta^2, 1 - \frac{p}{2}\right\} \cdot \Phi^k < \Phi^k. \tag{6}$$

In contrast to existing optimization algorithms whose convergence results typically rely on a single indicator, Thm.1 employs two accuracy indicators, $\Phi_k$ and $\min\{||\nabla f(w^k)||, ||x^k - x^\star||\}$. Specifically, for any given tolerance $e$, the Lyapunov function $\Phi^k$ decreases linearly until $\min\{||\nabla f(w^k)||, ||x^k - x^\star||\} \leq e$. Moreover, we emphasize that in practical implementations, achieving $||x^k - x^\star|| \leq e$ does not require choose a large $C_2$ at the beginning of the algorithm. Instead, we can gradually increase $C_2$ during the convergence process to enforce convergence, thus avoiding the injection of excessive noise at the early stage. We now turn to the convergence analysis in the nonconvex setting:

**Theorem 2** (Nonconvex Case). *Suppose Assumptions 3.1, 3.2, 3.4, 3.5 hold. For any given constant DP noise multipliers $\sigma_1, \sigma_2$ and $C_1 > 1, C_2 > 1$, let $\{x^k\}_{k=0}^T$ and $\{w^k\}_{k=0}^T$ be generated by Alg.3 with $\eta = \sqrt{\frac{2(f(x^0)-f(x^\star))}{TL\tilde{G}}} = O(\frac{1}{\sqrt{T}})$. Then,*

$$\frac{1}{T}\sum_{k=1}^T \mathbb{E}\left[\lambda_1^k||\nabla f(x^k)||^2 + \lambda_2^k||\nabla f(x^k)|| \cdot ||\nabla f(w^k)|| + \lambda_3^k||\nabla f(x^k)|| \cdot ||\nabla f(x^k) - \nabla f(w^k)||\right]$$

$$\tag{7}$$

$$\leq 2\sqrt{\frac{(f(x^0) - f(x^\star))L\tilde{G}}{2T}} + \frac{1}{T}\sum_{k=1}^T \mathbb{E}\left[\lambda_4^k \cdot 3\tau||\nabla f(x^k)||\right].$$

*Here, $\tilde{G} := 4G^2(4C_1^2(d\sigma_1^2+1) + C_2^2(d\sigma_2^2+1))$, $d$ denotes the model size, and for each $k$:*

$$\lambda_1^k := 1 - \frac{1}{3}(1 - \mathbb{P}^k)(2\sqrt{1 - \mathbb{P}_1^k} + \sqrt{1 - \mathbb{P}_2^k}), \quad \lambda_2^k := (1 - \mathbb{P}_2^k)(C_2 - 1), \quad \lambda_3^k := (1 - \mathbb{P}_1^k)(C_1 - 1)$$

$$\lambda_4^k := \frac{1}{3}\mathbb{P}^k(2\sqrt{1 - \mathbb{P}_1^k} + \sqrt{1 - \mathbb{P}_2^k}), \quad \mathbb{P}^k := \Pr\left(||\nabla f(x^k)|| \leq 3\tau \mid x^{k-1}\right),$$

$$\mathbb{P}_1^k := \mathbb{E}_k\left[1_{\{||\nabla f_i(x^k) - \nabla f_i(w^k)|| \leq C_{1k}\}}\right], \quad \mathbb{P}_2^k := \mathbb{E}_k\left[1_{\{||\nabla f_i(w^k)|| \leq C_{2k}\}}\right].$$

Since $\lambda_1^k \to 0$ and $\lambda_2^k \to 0$ require $\mathbb{P}_2^k \to 0$ and $\mathbb{P}_2^k \to 1$ respectively, $\lambda_1^k$ and $\lambda_2^k$ can not be zero simultaneously. Thus, Thm.2 effectively characterizes convergence. It is worth noting that (7) is obtained by a piecewise discussion of $||\nabla f(x^k)||$ (more detail see Appendix C.3): On the one hand, when $||\nabla f(x^k)|| \geq 3\tau$, the iteration exhibits strict descent, which guarantees that DP-C4 converges to the region $||\nabla f(x^k)|| < 3\tau$. On the other hand, when $||\nabla f(x^k)|| < 3\tau$, due to clipping bias, the right hand side introduces an optimization bias term $\frac{1}{T}\sum 3\tau\mathbb{E}[\lambda_4^k||\nabla f(x^k)||]$. However, Thm.2 differs from prior work in the following aspects: (i) compared with a fixed clipping bias at the constant scale proportional to $\tau$ (Xiao et al., 2023), the optimization bias term in (7) is proportional to $||\nabla f(x^k)||$, which implies a gradually vanishing clipping bias; (ii) in the $k$-th iteration, the last two terms on the left hand side of (7) also contribute to reducing the optimization bias. By employing the Cauchy-Schwarz inequality and setting $C_1 = (C_2 - 1)\frac{1 - \mathbb{P}_2^k}{1 - \mathbb{P}_1^k} + 1$, we obtain:

$$3\lambda_4^k\tau||\nabla f(x^k)|| - \lambda_2^k||\nabla f(x^k)|| \cdot ||\nabla f(w^k)|| - \lambda_3^k||\nabla f(x^k)|| \cdot ||\nabla f(x^k) - \nabla f(w^k)||$$

$$\leq 3\lambda_4^k\tau||\nabla f(x^k)|| - \lambda_3^k||\nabla f(x^k)||^2 \leq \frac{9(\lambda_4^k)^2\tau^2}{4(1 - \mathbb{P}_2^k)(C_2 - 1)} \xrightarrow{C_2 \to \infty} 0.$$

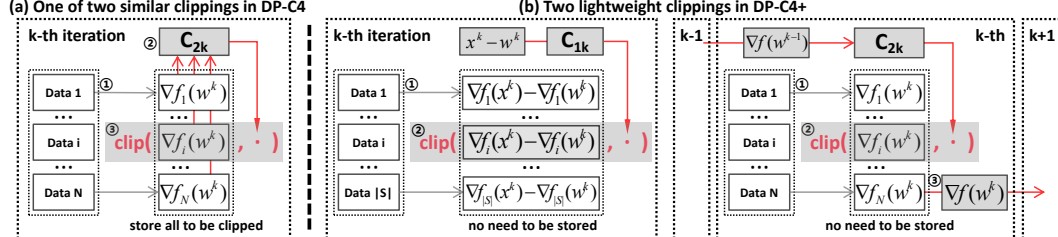

Figure 1: Workflow of DP-C4 and DP-C4$^+$

This indicates that by gradually and slowly increasing $C_2$ during the iteration, together with the decaying step size, the algorithm can converge to arbitrary accuracy.

## 3.5 PRIVACY ANALYSIS

In this subsection, we present the privacy guarantee of DP-C4. Since DP-C4 independently clips two components at each iteration, we carefully allocate the privacy budget between them and leverage Rényi differential privacy (RDP) (Mironov, 2017) to quantify the required noise magnitude at each step. Specifically, we have the following theorem:

**Theorem 3** (Noise Level). *Let $\theta = \frac{|D|^2}{|S|^2}$ and $\sigma^2 = \frac{4T(2\log(1/\delta)+\epsilon)}{|S|^2\epsilon^2}$. There exist $\sigma_1^2, \sigma_2^2$ defined in Alg.3 that guarantee $(\epsilon, \delta)$-DP of running DP-C4 with routine 1-4 for $T$ iterations:*

$$(\sigma_1^2, \sigma_2^2)_{R_{1\&2}} = \left((1+\sqrt{\tfrac{p}{\theta}})\sigma^2, (\tfrac{p}{\theta}+\sqrt{\tfrac{p}{\theta}})\sigma^2\right), \ (\sigma_1^2, \sigma_2^2)_{R_{3\&4}} = \left((1-p+\sqrt{\tfrac{p(1-p)}{\theta}})\sigma^2, (\tfrac{p}{\theta}+\sqrt{\tfrac{p(1-p)}{\theta}})\sigma^2\right).$$

It follows directly that, $(\sigma_1^2 + \sigma_2^2)_{R_{1\&2}} = (1+\sqrt{\tfrac{p}{\theta}})^2 \cdot \sigma^2 \approx \sigma^2$, $(\sigma_1^2 + \sigma_2^2)_{R_{3\&4}} = (\sqrt{\tfrac{p}{\theta}}+\sqrt{1-p})^2 \cdot \sigma^2 = (\sqrt{\tfrac{p}{\theta}}+1-\tfrac{p}{2}-\tfrac{p^2}{8}-O(p^3))^2 \cdot \sigma^2$. In practice, we choose the update probability $p = \frac{2|S|}{|D|} = \frac{2}{\sqrt{\theta}}$, guided by the probability $p$ is typically related to $\frac{|S|}{|D|}$ in SVRG (Kovalev et al., 2020). At the $k$-th iteration, the upper bound of the total noise variance $C_{1k}^2\sigma_1^2 + C_{2k}^2\sigma_2^2$ is as follows:

$$(C_{1k}^2\sigma_1^2 + C_{2k}^2\sigma_2^2)_{R_{3\&4}} \le (\sigma_1^2 + \sigma_2^2)\max\{C_{1k}^2, C_{2k}^2\} \le \sigma^2(1-O(p^3))^2\max\{C_{1k}^2, C_{2k}^2\}$$
$$< \sigma^2\max\{C_{1k}^2, C_{2k}^2\} = \sigma^2\min\{C^2, \max\{C_1^2||\nabla f_S(x^k)-\nabla f_S(w^k)||^2, C_2^2||\nabla f(w^k)||^2\}\}$$

It should be noted that $\sigma^2$ is exactly the noise multiplier in DP-SGD with a mini-batch size $|S|$. That is, for the same $C$, the total noise multiplier in DP-C4 is approximately the same as that in DP-SGD, with noise variance further decaying through $C_{1k}^2$ and $C_{2k}^2$.

## 4 DP-C4$^+$: A MEMORY-EFFICIENT EXTENSION OF DP-C4

In this section, we aim to reduce the memory burden of DP-C4, which currently requires storing every sampled gradient. This is because the gradients are first aggregated to determine the clipping thresholds, and then each is clipped individually. Note that the gradient difference $||\nabla f_i(x^k) - \nabla f_i(w^k)||$ can be bounded by $L||x^k - w^k||$ under the $L$-smoothness assumption, which tends to $0$ as $x^k, w^k \to x^\star$. In addition, since the anchor term is update only with probability $p$, it incurs limited memory overhead. Furthermore, rather than using $w^k$ to determine $C_{2k}$, we consider using the previous iterate $w^{k-1}$, which leads to $C_{2k}$ can be computed in advance. Specifically, we make the following substitutions in DP-C4:

$$C_{1k} = \min\left\{C, C_1 \cdot ||x^k - w^k||\right\}, \quad C_{2k} = \min\left\{C, C_2 \cdot ||\nabla f(w^{k-1})||\right\},$$

which we refered to as DP-C4$^+$. The workflow of DP-C4$^{(+)}$ is presented in Figure 1. Notably, the clipping thresholds of DP-C4$^+$ do not depend on the gradients of the current iterate and can

be precomputed. This design removes the need to store all gradients involved in the computation. On the one hand, DP-C4$^+$ does not violate our design principles and thus retains the properties of consistently-vanishing error, solution calibration, convergence guarantee, and DP guarantee. On the other hand, in practical deployment, to further reduce computational overhead, we often select a large batch size $|D'| >> |S|$ instead of the full dataset size $|D|$ as the anchor batch. This choice also helps to reduce the solution bias and improve utility, since it often holds that $||\frac{1}{|D'|} \sum_{i \in D'} \nabla f_i(x^\star)|| < ||\frac{1}{|S|} \sum_{i \in S} \nabla f_i(x^\star)||$. Due to the space limitation, we provide the pseudocode of DP-C4$^+$ and a detailed description of its properties in Appendix A.

## 5 NUMERICAL EXPERIMENTS

We conducted extensive experiments to demonstrate the advantages of DP-C4$^{(+)}$. Specifically, we evaluated our method on Mushroom (mus, 1981), Mnist (Deng, 2012), Cifar-10, Cifar-100 (Krizhevsky et al., 2009), IMDb (Maas et al., 2011), and GLUE (Wang et al., 2018) datasets, comparing against both related baselines and state-of-the-art methods, namely DP-SGD (Abadi et al., 2016), DP-SVRG (Lee, 2017), and DiceSGD (Zhang et al., 2023b). In addition, we conducted a series of ablation studies on CIFAR-10 to systematically evaluate the effects of the clipping thresholds $C_1, C_2$, the overall clipping threshold $C$, different update routines, varying large-batch sizes, and update probabilities. Due to space constraints, the detailed results and discussions are provided in Appendix E.

Table 1: Test accuracy of different methods on different datasets.

| Method | SVM | CV Tasks | | | NLP Tasks | |
|---|---|---|---|---|---|---|
| | Mushroom | Mnist | Cifar-10 | Cifar-100 | IMDb | GLUE SST-2 |
| DP-SGD | 87.48 | 96.26 | 53.05 | 37.04 | 76.99 | 75.23 |
| DP-SVRG | 77.13 | 95.79 | 51.81 | 31.08 | 74.10 | 72.71 |
| DiceSGD | 90.65 | 97.02 | 60.24 | 40.73 | 78.19 | 78.71 |
| DP-C4 | **91.76** | **96.93** | **61.89** | **43.46** | **80.13** | **81.31** |
| DP-C4$^+$ | **96.98** | **97.16** | **64.50** | **43.12** | **81.23** | **82.24** |

In our main experiments, we set the clipping thresholds to 1 for all methods, including $C$, $C_1$, and $C_2$ in DP-C4$^{(+)}$. The step size $\eta$ was tuned via grid search over $\{0.1, 0.05, 0.025, 0.0125\}$, and we report the best-performing results. For all mini-batches, we use a batch size of $|S| = 256$. In DP-C4$^{(+)}$ and DP-SVRG, we further set the large batch size to $|D'| = 4096$, and the update probability to $p = \frac{2|S|}{|D'|} = 0.125$. For the SVM task, we set the privacy parameters to $(\epsilon, \delta) = (1, 10^{-5})$, train for 50 epochs, and employ a logistic regression model on the Mushroom dataset. For image classification tasks, we set $(\epsilon, \delta) = (5, 10^{-5})$, train for 100 epochs, and adopt LeNet (LeCun et al., 2002) on Mnist, and ResNet20 (He et al., 2016) on CIFAR-10 and CIFAR-100. For NLP tasks, we set $(\epsilon, \delta) = (2, 10^{-5})$, train for 50 epochs, and adopt a GRU-RNN (Cho et al., 2014) on both IMDb and GLUE. The results are summarized in Table 1, where we observe that DP-C4$^{(+)}$ consistently outperforms the baselines across SVM, image classification, and NLP tasks.

## 6 CONCLUSION

In this work, we proposed DP-C4 and its variant DP-C4$^+$, which reconstruct the update rule and the clipping scheme of DP optimization to ensure that clipping bias and noise variance asymptotically vanish, thereby eliminating the solution bias inherent in existing methods. We established convergence guarantees by constructing a Lyapunov function under the $\mu$-strongly convex setting and identifying a vanishing bias term in the general non-convex case, offering a novel perspective on DP optimization analysis. On the privacy side, we designed a structure-aware budget allocation tailored to the coupled clipping framework, leading to general $(\epsilon, \delta)$-DP guarantees. Experiments on SVM, image classification, and NLP tasks demonstrate that DP-C4$^{(+)}$ consistently achieves superior privacy-utility trade-offs, underscoring its promise for practical deployment.

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

## A  DETAILS OF DP-C4$^+$

We present the pseudocode of DP-C4$^+$ in Alg. 4. As can be seen, the main difference from DP-C4 lies in the computation of the thresholds (Line 6&7). Moreover, at the beginning of the algorithm, we set the clipping threshold as $C_{2k} = C$ to accommodate the initialization at $k = 0$ (Line 3). Subsequently, we examine in detail the properties of DP-C4$^+$ as previously outlined.

---

**Algorithm 4** DP-C4$^+$

---

1: **Input:** Dataset $\mathcal{D}$, learning rate $\eta$, clipping bounds $C, C_1, C_2$, noise scales $\sigma_1, \sigma_2$, total steps $T$, anchor update probability $p$
2: **Output:** Model parameters $x^T$ satisfying $(\varepsilon, \delta)$-DP
3: **Initialize:** $x^0 = w^0 \in \mathbb{R}^d$, let $C_2||\nabla f(w^{-1})|| := C$
4: **for** $k = 0$ to $T - 1$ **do**
5:     Sample $S \subseteq D$
6:     $C_{1k} \leftarrow \min(C, C_1||x^k - w^k||)$                    {Pointwise coupled threshold}
7:     $C_{2k} \leftarrow \min(C, C_2||\nabla f(w^{k-1})||)$                    {Shifted anchor threshold}
8:     $g_1^k \leftarrow \frac{1}{|S|}\sum_{i\in S} \texttt{clip}(\nabla f_i(x^k) - \nabla f_i(w^k), C_{1k})$                    {Coupled term}
9:     $g_2^k \leftarrow \frac{1}{|D|}\sum_{i\in D} \texttt{clip}(\nabla f_i(w^k), C_{2k})$                    {Anchor term}
10:     $n_1^k \sim \mathcal{N}(0, \sigma_1^2 C_{1k}^2 I), \quad n_2^k \sim \mathcal{N}(0, \sigma_2^2 C_{2k}^2 I)$                    {Sample DP noise}
11:     $\tilde{g}^k \leftarrow g_1^k + g_2^k + n_1^k + n_2^k$                    {Add noise}
12:     $x^{k+1} \leftarrow x^k - \eta \cdot \tilde{g}^k$                    {Update model}
13:     Four alternative anchor update routines:
14:     $w^{k+1} \leftarrow \begin{cases} x^k, & \text{with probability } p \\ w^k, & \text{with probability } 1-p \end{cases}$                    {Update anchor (Routine 1)}
15:     $w^{k+1} \leftarrow \begin{cases} x^k, & k = 1 \ (mod \ [1/p]) \\ w^k, & k \neq 1 \ (mod \ [1/p]) \end{cases}$                    {Update anchor (Routine 2)}
16:     $w^{k+1} \leftarrow \begin{cases} x^{k+1}, & \text{with probability } p \\ w^k, & \text{with probability } 1-p \end{cases}$                    {Update anchor (Routine 3)}
17:     $w^{k+1} \leftarrow \begin{cases} x^{k+1}, & k = 1 \ (mod \ [1/p]) \\ w^k, & k \neq 1 \ (mod \ [1/p]) \end{cases}$                    {Update anchor (Routine 4)}
18: **end for**

---

**Consistently-vanishing Error**   We point out that the clipping bias and noise variance of DP-C4$^+$ also vanish. Specifically, continuing with the notation from Section 3.1, we have:

$$B_k + V_k \leq \frac{|I_2^k|}{|S|^2}\sum_{i\in I_2^k} \left[||\nabla f_S(x^k) - \nabla f_S(w^k)|| - C_1||x^k - w^k||\right]^2 + \sigma_1^2 C_1^2||x^k - w^k||^2$$

$$+ ||\frac{1}{|D|}\sum_{i\in D} clip(\nabla f_i(w^k), C_2||\nabla f(w^{k-1})||) - \nabla f(w^k)||^2 + \sigma_2^2 C_2^2||\nabla f(w^{k-1})||^2 \tag{8}$$

$$\leq (\frac{|I_2^k|^2(L-C_1)^2}{|S|^2} + \sigma_1^2 C_1^2)||x^k - w^k||^2 + (\sigma_2^2 + 2)C_2^2||\nabla f(w^{k-1})||^2 + 2||\nabla f(w^k)||^2 \xrightarrow{x^k, w^k \to x^\star} 0$$

**Solution Calibration**   Similarly, $(\tilde{x}, \tilde{w})$ is a fixed point of DP-C4$^+$ if and only if it is a solution to the original optimization problem. Analogously to (5), it must satisfy $C_1||\tilde{x} - \tilde{w}||_2 =$

$C_2\|\nabla f(\tilde{w})\|_2 = 0$, which implies $\tilde{x} = \tilde{w} = x^\star$. Specifically, at a potential fixed point $(\tilde{x}, \tilde{w})$, the iterative scheme of DP-C4$^+$ yields:

$$\begin{cases} \tilde{x} = x^{k+1} = x^k - \eta \tilde{g}^k = \tilde{x} - \eta \tilde{g}^k, \\ \tilde{w} = w^{k+1} = \begin{cases} x^k = \tilde{x}, & \text{with probability } p \\ w^k, & \text{with probability } 1-p \end{cases} \end{cases}.$$

From the iterative scheme of DP-C4$^+$ (Alg. 4), we obtain:

$$\begin{cases} \frac{1}{|S|}\sum_{i\in S}\text{clip}\left(\nabla f_i(\tilde{x}) - \nabla f_i(\tilde{w}), C_{1k}\right) + \frac{1}{|D|}\sum_{i\in D}\text{clip}(\nabla f_i(\tilde{w}), C_{2k}) + \mathbf{n}_1^k + \mathbf{n}_2^k = 0, \\ \tilde{w} = w^{k+1} = \begin{cases} x^k = \tilde{x}, & \text{with probability } p, \\ w^k, & \text{with probability } 1-p, \end{cases} \\ \mathbf{n}_1^k \sim \mathcal{N}(0, \sigma_1^2 C_{1k}^2 I), \quad \mathbf{n}_2^k \sim \mathcal{N}(0, \sigma_2^2 C_{2k}^2 I), \\ C_{1k} = \min(C, C_1\|\tilde{x} - \tilde{w}\|_2), \\ C_{2k} = \min(C, C_2\|\frac{1}{|D|}\sum_{i=1}^{|D|}\nabla f_i(\tilde{w})\|_2). \end{cases} \quad (9)$$

On the one hand, at a fixed point, (9) must be satisfied. This enforces that the variance of the injected noise vanishes almost surely, i.e., $C_{1k} = C_{2k} = 0$, which in turn requires $\tilde{x} = \tilde{w} = x^\star$. On the other hand, substituting $(\tilde{x}, \tilde{w}) = (x^\star, x^\star)$ back into (9) shows that the equality indeed holds. Therefore, the fixed point of DP-C4$^+$ coincides with the optimal solution $x^\star$ of the original problem.

**Convergence Guarantee**  The convergence of DP-C4$^+$ is similar to DP-C4, we establish convergence guarantees for DP-C4$^+$ under both strongly convex and non-convex regimes, the proofs of which are uniformly presented in Appendix C:

**Theorem 4** (Strongly Convex Case). *Suppose Assumptions 3.1-3.5 hold. For any given $e > 0$ and constant DP noise multipliers $\sigma_1, \sigma_2$, let $\{x^k\}_{k\geq 0}$ and $\{w^k\}_{k\geq 0}$ be generated by Alg.4 with $\eta < \min\left\{\frac{\mu}{3N_1+A}, \frac{1}{2LN_2}\right\}, C_1 > 0, C_2 \geq \frac{\tau}{e} + 1$. When $\min\{\|\nabla f(w^k)\|, \|x^k - x^\star\|, \|w^k - x^\star\|\} > e$, define the Lyapunov function as:*

$$\Phi^k := \mathbb{E}\|x^k - x^\star\|^2 + \frac{2N_1\eta^2}{p}\mathbb{E}\|w^k - x^\star\|^2 + \frac{2N_2\eta^2}{p}D^k,$$

*where $D^k := \mathbb{E}\|\nabla f_i(w^k) - \nabla f_i(x^\star)\|^2, N_1 := 8C_1^2(d\sigma_1^2+1) + \frac{4\eta^2}{pe^2}G^2C_2^2(d\sigma_2^2+1), N_2 := 8C_2^2(d\sigma_2^2+1),$ $A := \frac{4G}{p\mu^2 e^2}[(pC_2+1)L - C_1]\sqrt{2C_1^2(d\sigma_1^2+1) + \mu^2 C_2^2(d\sigma_2^2+1)}$, and $d$ denotes the model size. Then,*

$$\Phi^{k+1} \leq \max\left\{1 - \mu\eta + (3N_1+A)\eta^2, 1 - \frac{p}{2}\right\} \cdot \Phi^k < \Phi^k. \quad (10)$$

**Theorem 5** (Nonconvex Case). *Suppose Assumptions 3.1, 3.2, 3.4, 3.5 hold. For any given constant DP noise multipliers $\sigma_1, \sigma_2$ and $C_1 > 1, C_2 > 1$, let $\{x^k\}_{k=0}^T$ and $\{w^k\}_{k=0}^T$ be generated by Alg.4 with $\eta = \sqrt{\frac{2(f(x^0) - f(x^\star))}{TL\tilde{G}(1+4G)}} = O(\frac{1}{\sqrt{T}})$. Then,*

$$\frac{1}{T}\sum_{k=1}^T \mathbb{E}\left[\lambda_1^k\|\nabla f(x^k)\|^2 + \lambda_2^k\|\nabla f(x^k)\| \cdot \|\nabla f(w^k)\| + \lambda_3^k\|\nabla f(x^k)\| \cdot \|\nabla f(x^k) - \nabla f(w^k)\|\right]$$

$$\leq 2\sqrt{\frac{(f(x^0) - f(x^\star))L\tilde{G}(1+4G)}{2T}} + \frac{1}{T}\sum_{k=1}^T \mathbb{E}\left[\lambda_4^k \cdot 3\tau\|\nabla f(x^k)\|\right]. \quad (11)$$

*Here, $\tilde{G} = 4C^2(d\sigma_1^2+1) + 4G^2C_2^2(d\sigma_2^2+1)$, $d$ denotes the model size, and for each $k$:*

$$\lambda_1^k := 1 - \frac{1}{3}(1 - \mathbb{P}^k)(2\sqrt{1 - \mathbb{P}_1^k} + \sqrt{1 - \mathbb{P}_2^k}), \quad \lambda_2^k := (1 - \mathbb{P}_2^k)(C_2 - 1), \quad \lambda_3^k := (1 - \mathbb{P}_1^k)(\frac{C_1}{L} - 1)$$

$$\lambda_4^k := \frac{1}{3}\mathbb{P}^k(2\sqrt{1 - \mathbb{P}_1^k} + \sqrt{1 - \mathbb{P}_2^k}), \quad \mathbb{P}^k := \Pr\left(\|\nabla f(x^k)\| \leq 3\tau \mid x^{k-1}\right),$$

$$\mathbb{P}_1^k := \mathbb{E}_k\left[1_{\{\|\nabla f_i(x^k) - \nabla f_i(w^k)\| \leq C_{1k}\}}\right], \quad \mathbb{P}_2^k := \mathbb{E}_k\left[1_{\{\|\nabla f_i(w^k)\| \leq C_{2k}\}}\right].$$

# B  ALGORITHM COMPARISON

In this section, we provide a detailed exposition of the fundamental distinction between DP-C4$^{(+)}$ and other algorithms (as an extension of Section 3.3), namely, the unique Solution-Calibrated Property that is exclusive to DP-C4$^{(+)}$ but absent in existing approaches.

In Section 3.3 and Appendix A, we have established the solution-calibrated property of DP-C4$^{(+)}$. In contrast, methods employing a constant clipping threshold (e.g., DP-SGD, DP-SVRG) do not admit fixed points, as the fixed-variance noise injected at each iteration continually disrupts equilibrium. Taking DP-SGD as an example, suppose it admits a fixed point $\tilde{x}$, we obtain:

$$\tilde{x} = x^{k+1} = x^k - \eta\tilde{g}^k = \tilde{x} - \eta\tilde{g}^k,$$

That is, $\tilde{g}^k = \frac{1}{|S_k|}\sum_{i\in S_k}\text{clip}(\nabla f_i(\tilde{x}), C) + \mathbf{n}^k = 0$. However, due to the stochasticity introduced by the noise in each iteration, this condition cannot be satisfied with probability 1. Consequently, DP-SGD does not admit a fixed point.

For other schemes where the clipping threshold decays (TD) to 0, the persistent gradient estimation noise at each iteration, and the gradual accumulation of clipping bias, combined with a mismatch between the decay rate of the threshold and the convergence speed, ensures that the fixed point is, with probability 1, not a solution to the original problem. Taking DP-SGD$^{TD}$ as an example, suppose it admits a fixed point $\tilde{x}$, we obtain:

$$\begin{cases} \tilde{g}^k = \frac{1}{|S_k|}\sum_{i\in S_k}\text{clip}(\nabla f_i(\tilde{x}), C_k) + \mathbf{n}^k = 0, \\ \mathbf{n}^k \sim \mathcal{N}(0, \sigma_1^2 C_k^2 I), \quad C_k \to 0 \end{cases}$$

We can observe that when the clipping threshold approaches zero (i.e., $C_k = 0$), the above equation is indeed satisfied, implying that DP-SGD with a decaying threshold admits a fixed point $\tilde{x}$. However, this fixed point arises from the elimination of the update due to the vanishing threshold, and therefore it does not guarantee that $\tilde{x} = x^\star$.

---

**Algorithm 5** DiceSGD (Zhang et al., 2023b)

---

1: **Input:** Dataset $\mathcal{D}$, learning rate $\eta$, clipping bounds $C_1, C_2$, noise scale $\sigma$, total steps $T$
2: **Output:** Model parameters $x^T$ satisfying $(\varepsilon, \delta)$-DP
3: **Initialize:** $e^0 = 0, x^0 \in \mathbb{R}^d$
4: **for** $k = 0, \ldots, T-1$ **do**
5:     Randomly draw minibatch $S$ from $D$
6:     $g^k = \frac{1}{|S|}\sum_{i\in S}\text{clip}\left(\nabla f_i(x^k), C_1\right) + \text{clip}\left(e^k, C_2\right)$
7:     $x^{k+1} = x^k - \eta(g^k + \mathbf{n}^k)$, where $\mathbf{n}^k \sim \mathcal{N}(0, \sigma^2(C_1^2 + C_2^2)\mathbf{I})$
8:     $e^{k+1} = e^k + \frac{1}{|S|}\sum_{i\in S}\nabla f_i(x^k) - g^k$
9: **end for**

---

Recently, the proposed DiceSGD (Zhang et al., 2023b) (Alg 9) eliminates the bias in each iteration *in expectation*. Therefore, in the sense of ignoring the injected noise and sampling randomness (i.e., in the full-expectation sense), it possesses a similar property. Assume that $(\tilde{x}, \tilde{e})$ is a fixed point of DiceSGD, then we have:

$$\begin{cases} \mathbb{E}[\tilde{x}] = \mathbb{E}[\tilde{x}] - \eta\mathbb{E}[g^k + \mathbf{n}^k] = \mathbb{E}[\tilde{x}] - \eta\mathbb{E}[g^k], \\ \mathbb{E}[\tilde{e}] = \mathbb{E}[\tilde{e}] + \mathbb{E}\left[\frac{1}{|S|}\sum_{i\in S}\nabla f_i(\mathbf{x}) - g^k\right] = \mathbb{E}[\tilde{e}] + \frac{1}{N}\sum_{i=1}^N\nabla f_i(\tilde{x}) - \mathbb{E}[g^k]. \end{cases} \quad (12)$$

We can verify that $(\tilde{x}, \tilde{e}) = (x^\star, 0)$ is indeed a solution to (12), implying that, *in full-expectation sense*, the fixed point of DiceSGD coincides with the solution of the original problem. However, as discussed earlier, the randomness introduced by noise and sampling can disrupt this balance at any iteration, causing the iterates to deviate from the true solution.

Specifically, Table 2 summarizes the solution calibration property of different methods under both noise and sampling stochasticity, where the symbols $-$, $\checkmark$ and $\times$ respectively denote: no fixed point exists, the fixed point is (not) a solution to the problem. $\mathbb{E}_{full}$, $\mathbb{E}_{noise}$, $\mathbb{E}_{sampling}$, no-$\mathbb{E}$ denote, respectively, *in the sense of full expectation*, *in the sense of expectation over noise*, *in the sense of expectation over sampling*, and *taking into account all sources of randomness*.

Table 2: Algorithm Comparison on Solution-Calibrated Property.

| Type of $\mathbb{E}$ \ Method | DP-SGD$^{(TD)}$ | DP-SVRG$^{(TD)}$ | DiceSGD | DP-C4$^{(+)}$ |
|---|---|---|---|---|
| $\mathbb{E}_{full}$ | -($\times$) | -($\times$) | $\checkmark$ | $\checkmark$ |
| $\mathbb{E}_{noise}$ | -($\times$) | -($\times$) | - | $\checkmark$ |
| $\mathbb{E}_{sampling}$ | -($\times$) | -($\times$) | - | $\checkmark$ |
| no-$\mathbb{E}$ | -($\times$) | -($\times$) | - | $\checkmark$ |

## C  PROOFS OF CONVERGENCE ANALYSIS

In this section, we present the detailed proofs of the convergence results of DP-C4 and DP-C4$^+$, i.e., Lemma.1, Thm.1-2, and Thm.4-5. It is worth noting that the proof techniques for DP-C4$^+$ closely follow those of DP-C4, and we mainly highlight the differences for clarity.

### C.1  PROOF OF LEMMA 1

According to the definition of the clipping bias $B^k$ (Section 3.1), we can directly obtain:

$$
\begin{aligned}
B^k &= \|\frac{1}{|S|}\sum_{i\in S} clip(h_i(x^k), C(\{h_i(x^k)\}_{i\in S})) - \frac{1}{|S|}\sum_{i\in S} h_i(x^k)\|_2^2 \\
&\stackrel{(a)}{=} \|\frac{1}{|S|}\sum_{i\in I_1^k} h_i(x^k) + \frac{1}{|S|}\sum_{i\in I_2^k} \frac{C(\{h_i(x^k)\}_{i\in S})}{\|h_i(x^k)\|_2}\cdot h_i(x^k) \\
&\quad - \frac{1}{|S|}\sum_{i\in I_1^k} h_i(x^k) - \frac{1}{|S|}\sum_{i\in I_2^k} h_i(x^k)\|_2^2 \\
&= \|\frac{1}{|S|}\sum_{i\in I_2^k}(\frac{C(\{h_i(x^k)\}_{i\in S})}{\|h_i(x^k)\|_2} - 1)\cdot h_i(x^k)\|_2^2 \\
&= \frac{1}{|S|^2}\|\sum_{i\in I_2^k}\underbrace{(C(\{h_i(x^k)\}_{i\in S}) - \|h_i(x^k)\|_2)}_{\leq 0}\cdot \frac{h_i(x^k)}{\|h_i(x^k)\|}\|_2^2 \\
&\leq \frac{1}{|S|^2}\Big(\sum_{i\in I_2^k}(\|h_i(x^k)\|_2 - C(\{h_i(x^k)\}_{i\in S}))\cdot \frac{\|h_i(x^k)\|}{\|h_i(x^k)\|}\Big)^2 \\
&= \frac{1}{|S|^2}\Big[\sum_{i\in I_2^k}(\|h_i(x^k)\|_2 - C(\{h_i(x^k)\}_{i\in S}))\Big]^2 \\
&\stackrel{(b)}{\leq} \frac{|I_2^k|}{|S|^2}\sum_{i\in I_2^k}\Big[\|h_i(x^k)\|_2 - C(\{h_i(x^k)\}_{i\in S})\Big]^2
\end{aligned}
\tag{13}
$$

### C.2  PROOF OF THEOREM 1

For the strongly convex case of DP-C4, our goal is to construct a Lyapunov function under appropriately chosen clipping coefficients. We first examine a potential term in the Lyapunov function of the system, namely $\mathbb{E}\|x^k - x^\star\|^2$. Combining this with the update rule of DP-C4, and denoting the clipping biases as $b_1^k := \frac{1}{|S|}\sum_{i\in S}\text{clip}(\Delta_i^k, C_1\|\Delta_S^k\|) - \Delta_S^k$ and $b_2^k :=$

$\frac{1}{|D|}\sum_{i\in D}\text{clip}(\nabla f_i(w^k), C_2||\nabla f(w^k)||) - \nabla f(w^k)$, we obtain:

$$
\begin{aligned}
\mathbb{E}_k||x^{k+1}-x^\star||^2 &= \mathbb{E}_k||x^k-x^\star-\eta\tilde{g}^k||^2 \\
&= ||x^k-x^\star||^2+\mathbb{E}_k[2\eta\langle\tilde{g}^k, x^\star-x^k\rangle]+\eta^2\mathbb{E}_k||\tilde{g}^k||^2 \\
&\le ||x^k-x^\star||^2+2\eta\mathbb{E}_k\langle\Delta_S^k+\nabla f(w^k)+b_1^k+b_2^k+\mathrm{n}_1^k+\mathrm{n}_2^k, x^\star-x^k\rangle+\eta^2\mathbb{E}_k||\tilde{g}^k||^2 \\
&= ||x^k-x^\star||^2+2\eta\langle\nabla f(x^k), x^\star-x^k\rangle+2\eta\mathbb{E}_k\langle b_1^k+b_2^k+\mathrm{n}_1^k+\mathrm{n}_2^k, x^\star-x^k\rangle+\eta^2\mathbb{E}_k||\tilde{g}^k||^2 \\
&\overset{(a)}{\le} ||x^k-x^\star||^2+2\eta\underbrace{(f^\star-f(x^k)-\frac{\mu}{2}||x^k-x^\star||^2)}_{\mu-strongly\ convex}+2\eta\mathbb{E}_k\langle b_1^k+b_2^k, x^\star-x^k\rangle+\eta^2\mathbb{E}_k||\tilde{g}^k||^2 \\
&= ||x^k-x^\star||^2(1-\eta\mu)+2\eta(f^\star-f(x^k))+2\eta\mathbb{E}_k\langle b_1^k+b_2^k, x^\star-x^k\rangle+\eta^2\mathbb{E}_k||\tilde{g}^k||^2
\end{aligned}
$$

(14)

Here, (a) follows from the $\mu$-strong convexity property, together with the fact that $\mathbb{E}_k[\mathrm{n}_1^k] = \mathbb{E}_k[\mathrm{n}_2^k] = 0$. Next, we derive upper bounds for the last two terms in the above expression. Specifically, we begin by analyzing the upper bound of $\mathbb{E}_k||\tilde{g}^k||^2$, for which we have:

$$
\begin{aligned}
\mathbb{E}_k||\tilde{g}^k||^2 &= \mathbb{E}_k[||\frac{1}{|S|}\sum_{i\in S}clip(\nabla f_i(x^k)-\nabla f_i(w^k), C_1||\nabla f_S(x^k)-\nabla f_S(w^k)||) \\
&\quad +\frac{1}{N}\sum_{i\in D}clip(\nabla f_i(w^k), C_2||\nabla f(w^k)||)+\mathrm{n}_1^k+\mathrm{n}_2^k||^2] \\
&\overset{(a)}{\le} 4\mathbb{E}_k||\frac{1}{|S|}\sum_{i\in S}clip(\nabla f_i(x^k)-\nabla f_i(w^k), C_1||\nabla f_S(x^k)-\nabla f_S(w^k)||)||^2 \\
&\quad +4||\frac{1}{|D|}\sum_{i\in D}clip(\nabla f_i(w^k), C_2||\nabla f(w^k)||)||^2 \\
&\quad +4dL^2\sigma_1^2 C_1^2||x^k-w^k||^2+4d\sigma_2^2 C_2^2||\nabla f(w^k)||^2 \\
&\overset{(b)}{\le} 4L^2 C_1^2(d\sigma_1^2+1)||x^k-x^\star+x^\star-w^k||^2+4C_2^2(d\sigma_2^2+1)||\nabla f(w^k)||^2 \\
&\overset{(c)}{\le} \underbrace{8L^2 C_1^2(d\sigma_1^2+1)}_{:=N_1}||x^k-x^\star||^2+\underbrace{8L^2 C_1^2(d\sigma_1^2+1)}_{:=N_1}||w^k-x^\star||^2 \\
&\quad +\underbrace{4C_2^2(d\sigma_2^2+1)}_{:=N_2}\cdot\underbrace{\frac{1}{|D|}\sum_{i\in[|D|]}||\nabla f_i(w^k)-\nabla f_i(x^\star)||^2}_{:=D^k}
\end{aligned}
$$

(15)

Here, $d$ denotes the model size. Inequality (a) follows from the Cauchy–Schwarz inequality and the $L$-smoothness property applied to the noise term $||\mathrm{n}_1^k||^2$; (b) applies the $L$-smoothness property to the first clipping term; and (c) uses the Cauchy–Schwarz inequality along with the convexity of the squared $\ell_2$-norm, i.e., $||\mathbb{E}[X]||^2 \le \mathbb{E}[||X||^2]$. For $\mathbb{E}_k[||\tilde{g}^k||]$, we have:

$$
\begin{aligned}
\mathbb{E}_k||\tilde{g}^k|| &\overset{(a)}{\le} \sqrt{\mathbb{E}_k||\tilde{g}^k||^2} \\
&\overset{(b)}{\le} (4L^2 C_1^2(d\sigma_1^2+1)||x^k-w^k||^2+4C_2^2(d\sigma_2^2+1)||\nabla f(w^k)||^2)^{\frac{1}{2}} \\
&\overset{(c)}{\le} (\frac{4L^2 C_1^2(d\sigma_1^2+1)}{\mu^2}||\nabla f(x^k)-\nabla f(w^k)||^2+4C_2^2(d\sigma_2^2+1)||\nabla f(w^k)||^2)^{\frac{1}{2}} \\
&\overset{(d)}{\le} (\frac{16L^2 C_1^2(d\sigma_1^2+1)G^2}{\mu^2} + 4C_2^2(d\sigma_2^2+1)G^2)^{\frac{1}{2}} \\
&= \frac{2G}{\mu}\sqrt{4L^2 C_1^2(d\sigma_1^2+1) + \mu^2 C_2^2(d\sigma_2^2+1)} := \tilde{G}
\end{aligned}
$$

(16)

For any precision $e > 0$, when $||\nabla f(w)||, ||x^k-x^\star|| > e$, we define the unclipped and clipped sample sets for the first clipping as $J_1^k := \{j : ||\Delta_j^k|| \le C_{1k}\}$ and $J_2^k := \{j : ||\Delta_j^k|| > C_{1k}\}$,

and those induced by the second clipping as $I_1^k := \{i : ||\nabla f_i(w^k)|| \leq C_{2k}\}$ and $I_2^k := \{i : ||\nabla f_i(w^k)|| > C_{2k}\}$. By choosing $C_1 > 0$ and $C_2 \geq \frac{\tau}{e} + 1$, we have:

$$b_1^k = \frac{1}{|S|}\sum_{i \in S} clip(\Delta_i^k, C_{1k}) - \Delta_S^k = \frac{1}{|S|}(\sum_{i \in J_1^k} \Delta_i^k + \sum_{i \in J_2^k} \frac{C_{1k}}{||\Delta_i^k||}\Delta_i^k) - \Delta_S^k$$

$$= \frac{1}{|S|}\sum_{i \in J_2^k}(\frac{C_{1k}}{||\Delta_i^k||} - 1)\cdot\Delta_i^k = \frac{1}{|S|}\sum_{i \in J_2^k}(C_1||\Delta_S^k|| - ||\Delta_i^k||)\cdot\frac{\Delta_i^k}{||\Delta_i^k||} \tag{17}$$

$$b_2^k = \frac{1}{|D|}\sum_{i \in D} clip(\nabla f_i(w^k), C_{2k}) - \nabla f(w^k) = \frac{1}{|D|}(\sum_{i \in I_1^k}\nabla f_i(w^k) + \sum_{i \in I_2^k}\frac{C_{2k}}{||\nabla f_i(w^k)||}\nabla f_i(w^k)) - \nabla f(w^k)$$

$$= \frac{1}{|D|}\sum_{i \in I_2^k}(\frac{C_{2k}}{||\nabla f_i(w^k)||} - 1)\cdot\nabla f_i(w^k) = \frac{1}{|D|}\sum_{i \in I_2^k}(C_2||\nabla f(w^k)|| - ||\nabla f_i(w^k)||)\cdot\frac{\nabla f_i(w^k)}{||\nabla f_i(w^k)||} \tag{18}$$

For the first clipping, we define the probability of an individual sample remaining unclipped as $\mathbb{P}_1^k := \mathbb{E}_k 1_{\{||\Delta_i^k|| \leq C_{1k}\}}$. Then, we have:

$$\mathbb{E}_k[\langle b_1^k, x^\star - x^k\rangle] = \mathbb{E}_k[\langle \frac{1}{|S|}\sum_{i \in J_2^k}\underbrace{(C_1||\Delta_S^k|| - ||\Delta_i^k||)}_{<0}\cdot\frac{\Delta_i^k}{||\Delta_i^k||}, x^\star - x^k\rangle]$$

$$\overset{(a)}{\leq} \mathbb{E}_k[\frac{1}{|S|}\sum_{i \in J_2^k}(||\Delta_i^k|| - C_1||\Delta_S^k||)\cdot||x^k - x^\star||]$$

$$\overset{(b)}{\leq} \mathbb{E}_k[\frac{1}{|S|}\sum_{i \in J_2^k}(L||x^k - w^k|| - C_1\mu||x^k - w^k||)\cdot||x^k - x^\star||] \tag{19}$$

$$\leq \mathbb{E}_k[\frac{1}{|S|}\sum_{i \in J_2^k}(L - C_1\mu)||x^k - w^k||\cdot||x^k - x^\star||]$$

$$= (1 - \mathbb{P}_1^k)(L - C_1\mu)||x^k - w^k||\cdot||x^k - x^\star||$$

$$\leq (L - C_1\mu)||x^k - w^k||\cdot||x^k - x^\star||$$

Here, (a) follows from the Cauchy–Schwarz inequality, (b) follows from the $L$-smoothness and $\mu$-strong convexity of the objective function. For the second clipping, similarly, we define $\mathbb{P}_2^k := \mathbb{E}_k 1_{\{||\nabla f_i(w^k)|| \leq C_{2k}\}}$, we have:

$$\mathbb{E}_k\langle b_2^k, x^\star - x^k\rangle = \mathbb{E}_k\langle\frac{1}{|D|}\sum_{i \in I_2^k}\underbrace{(C_2||\nabla f(w^k)|| - ||\nabla f_i(w^k)||)}_{<0}\cdot\frac{\nabla f_i(w^k)}{||\nabla f_i(w^k)||}, x^\star - x^k\rangle$$

$$\overset{(a)}{\leq} \mathbb{E}_k\frac{1}{|D|}\sum_{i \in I_2^k}(||\nabla f_i(w^k)|| - C_2||\nabla f(w^k)||)\cdot||x^k - x^\star||$$

$$\leq \mathbb{E}_k\frac{1}{|D|}\sum_{i \in I_2^k}(||\nabla f_i(w^k) - \nabla f(w^k) + \nabla f(w^k)|| - C_2||\nabla f(w^k)||)\cdot||x^k - x^\star||$$

$$\overset{(b)}{\leq} \mathbb{E}_k\frac{1}{|D|}\sum_{i \in I_2^k}(||\nabla f_i(w^k) - \nabla f(w^k)|| + ||\nabla f(w^k)|| - C_2||\nabla f(w^k)||)\cdot||x^k - x^\star|| \tag{20}$$

$$\overset{(c)}{\leq} \mathbb{E}_k\frac{1}{|D|}\sum_{i \in I_2^k}(\tau - (C_2 - 1)||\nabla f(w^k)||)\cdot||x^k - x^\star||$$

$$\overset{(d)}{\leq} \frac{1 - \mathbb{P}_2^k}{|D|}\underbrace{(\tau - (C_2 - 1)e)}_{\leq 0}\cdot||x^k - x^\star|| \leq 0$$

Here, (a) follows from the Cauchy–Schwarz inequality, (b) from the triangle inequality, (c) from Assumption 3.4, and (d) from our prescribed accuracy condition together with the choice of $C_2$.

To handle the above terms, we next examine the randomness in the $w^k$ iteration that arises both from the coin-flipping mechanism and from the stochasticity of the noise sampling. Owing to the independence of different sources of randomness, and in terms of the full expectation, we have:

$$\mathbb{E}[||x^k - w^k|| \cdot ||x^k - x^\star||]$$

$$\overset{(a)}{\leq} \mathbb{E}[||x^k - w^k|| \cdot \frac{||\nabla f(x^k) - \nabla f(x^\star)||}{\mu}]$$

$$= \frac{1}{\mu} \mathbb{E}[||x^k - w^k|| \cdot ||\nabla f(x^k)||]$$

$$\overset{(b)}{\leq} \frac{G}{\mu} \mathbb{E}||x^k - w^k||$$

$$\overset{(c)}{=} \frac{G}{\mu} \mathbb{E}[p||x^k - x^{k-1}|| + (1-p)||x^k - w^{k-1}||]$$

$$\overset{(d)}{\leq} \frac{G}{\mu} \mathbb{E}[||x^k - x^{k-1}|| + (1-p)||x^{k-1} - w^{k-1}||] \tag{21}$$

$$\leq \frac{G}{\mu} \mathbb{E}[(||x^k - x^{k-1}|| + (1-p)||x^{k-1} - x^{k-2}|| + (1-p)^2||x^{k-2} - x^{k-3}|| + \cdots$$

$$+ (1-p)^{k-1}||x^1 - x^0|| + p||w^0 - x^0||]$$

$$\leq \frac{G}{\mu} \mathbb{E}[\eta(||\tilde{g}^{k-1}|| + (1-p)||\tilde{g}^{k-2}|| + \cdots + (1-p)^k||\tilde{g}^0||]$$

$$\leq \frac{G}{\mu} \mathbb{E}[\eta\tilde{G}(1 + (1-p) + (1-p)^2 + \cdots + (1-p)^k]$$

$$\leq \eta \frac{G\tilde{G}}{p\mu e^2} \cdot e^2 \leq \eta \frac{G\tilde{G}}{p\mu e^2} \mathbb{E}||x^k - x^\star||^2$$

Here, (a) follows from the $\mu$-strong convexity property; (b) is due to Assumption 3.5; (c) comes from the iterative update rule of $w^k$; and (d) is obtained by applying the triangle inequality. Similarly, we also obtain the following results, which will be used in the subsequent proofs:

$$\mathbb{E}||x^{k-1} - w^{k-1}|| \cdot ||x^k - x^\star|| \leq \eta \frac{G\tilde{G}}{p\mu}$$

$$\mathbb{E}||w^k - w^{k-1}||^2 \leq \eta^2 \frac{\tilde{G}^2}{p} \tag{22}$$

$$\mathbb{E}||x^{k-1} - w^{k-1}|| \cdot ||\nabla f(x^k)|| \leq \eta \frac{G\tilde{G}}{p}$$

With these preparations in place, we are now ready to proceed. For notational simplicity, in (15) we define $D^k := \mathbb{E}||\nabla f_i(w^k) - \nabla f_i(x^\star)||^2$, $N_1 := 8L^2C_1^2(d\sigma_1^2 + 1)$, $N_2 := 4(d\sigma_2^2 + 1)C_2^2$, and $A := \frac{2G\tilde{G}}{p\mu e^2}(L - C_1\mu)$. Substituting (15)–(21) into (14), and taking the full expectation on both sides of (14), (15), and (19), we obtain:

$$\mathbb{E}||x^{k+1} - x^\star||^2 \leq (1 - \eta\mu + \eta^2(N_1 + A)\mathbb{E}||x^k - x^\star||^2 + \eta^2 N_1 \mathbb{E}||w^k - x^\star||^2$$
$$+ \eta^2 N_2 D^k - 2\eta(\mathbb{E}f(x^k) - f^\star) \tag{23}$$

We now consider the iterative update of $\{w^k\}_{k\in[T]}$ in DP-C4 (Line 13 in Alg.3). Since $w^k$ is updated with a certain probability, we have:

$$\mathbb{E}||w^{k+1} - x^\star||^2 = p\mathbb{E}||x^k - x^\star||^2 + (1-p)\mathbb{E}||w^k - x^\star||^2$$

$$D^{k+1} = (1-p)D^k + p\mathbb{E}||\nabla f_i(x^k) - \nabla f_i(x^\star)||^2 \tag{24}$$

$$\leq (1-p)D^k + 2Lp(\mathbb{E}f(x^k) - f^\star)$$

We define the Lyapunov function of DP-C4 as follows:

$$\Phi^k = \mathbb{E}||x^k - x^\star||^2 + \frac{2N_1\eta^2}{p}\mathbb{E}||w^k - x^\star||^2 + \frac{2N_2\eta^2}{p}D^k$$

$$= \mathbb{E}||x^k - x^\star||^2 + \frac{16L^2C_1^2(d\sigma_1^2+1)\eta^2}{p}\mathbb{E}||w^k - x^\star||^2 + \frac{8C_2^2(d\sigma_2^2+1)\eta^2}{p}D^k \tag{25}$$

Let $\eta < \min\left\{\frac{\mu}{3N_1+A}, \frac{1}{2LN_2}\right\}$. Then, we observe that:

$$
\begin{aligned}
\Phi^{k+1} &= \mathbb{E}||x^{k+1}-x^{\star}||^2 + \frac{2N_1\eta^2}{p}\mathbb{E}||w^{k+1}-x^{\star}||^2 + \frac{2N_2\eta^2}{p}D^{k+1} \\
&\leq (1-\mu\eta+(N_1+A)\eta^2+p\frac{2N_1\eta^2}{p})\mathbb{E}||x^k-x^{\star}||^2 + (N_1\eta^2+(1-p)\frac{2N_1\eta^2}{p})\mathbb{E}||w^k-x^{\star}||^2 \\
&\quad + (N_2\eta^2+(1-p)\frac{2N_2\eta^2}{p})D^k + \underbrace{(4LN_2\eta^2-2\eta)}_{<0}(\mathbb{E}f(x^k)-f^{\star}) \\
&= \underbrace{(1-\mu\eta+(3N_1+A)\eta^2)}_{<1}\mathbb{E}||x^k-x^{\star}||^2 + (1-\frac{p}{2})\frac{2N_1\eta^2}{p}\mathbb{E}||w^k-x^{\star}||^2 + (1-\frac{p}{2})\frac{2N_2\eta^2}{p}D^k
\end{aligned}
\tag{26}
$$

That is,

$$
\Phi^{k+1} \leq max\{\underbrace{1-\mu\eta+(3N_1+A)\eta^2}_{<1}, \underbrace{1-\frac{p}{2}}_{<1}\} \cdot \Phi^k < \Phi^k
\tag{27}
$$

### C.3 PROOF OF THEOREM 2

Unlike Thm.1, here we study the general convergence analysis in the non-convex setting without imposing stringent restrictions on the clipping coefficients $C_1$ and $C_2$. Therefore, we need to consider the clipping bias in a more refined manner. First, since $f(x)$ is $L$-smooth, we have:

$$
\begin{aligned}
f(x^{k+1}) - f(x^k) &\leq \langle\nabla f(x^k), x^{k+1}-x^k\rangle + \frac{L}{2}||x^{k+1}-x^k||^2 \\
&= -\eta\langle\nabla f(x^k), \tilde{g}^k\rangle + \frac{L\eta^2}{2}||\tilde{g}^k||^2.
\end{aligned}
\tag{28}
$$

Taking the expectation on both sides of the inequality, and let us define $g^k := \tilde{g}^k - n_1^k - n_2^k$, we obtain:

$$
\mathbb{E}_k[f(x^{k+1})] - f(x^k) \leq -\eta\mathbb{E}_k\langle\nabla f(x^k), g^k\rangle + \frac{L\eta^2}{2}\mathbb{E}_k[||g^k+n_1^k+n_2^k||^2]
\tag{29}
$$

Our current goal is to derive a lower bound for $\mathbb{E}_k\langle\nabla f(x^k), g^k\rangle$ and an upper bound for $\mathbb{E}_k[||g^k+n_1^k+n_2^k||^2]$. We first consider the upper bound of $\mathbb{E}_k[||g^k+n_1^k+n_2^k||^2]$. From (15), we have:

$$
\begin{aligned}
\mathbb{E}_k||\tilde{g}^k||^2 &\leq 4C_1^2||\nabla f_S(x^k)-\nabla f_S(w^k)||^2 + 4dC_1^2\sigma_1^2||\nabla f(x^k)-\nabla f(w^k)||^2 \\
&\quad + 4C_2^2(d\sigma_2^2+1)||\nabla f(w^k)||^2 \\
&\leq 8C_1^2(||\nabla f_S(x^k)||^2+||\nabla f_S(w^k)||^2) + 8dC_1^2\sigma_1^2(||\nabla f(x^k)||^2+||\nabla f(w^k)||^2) \\
&\quad + 4C_2^2(d\sigma_2^2+1)||\nabla f(w^k)||^2 \\
&\leq 4G^2(4C_1^2(d\sigma_1^2+1)+C_2^2(d\sigma_2^2+1)) := \tilde{G}'
\end{aligned}
\tag{30}
$$

We now discuss a lower bound for $\mathbb{E}_k\langle\nabla f(x^k), g^k\rangle$. Our approach is to use the gradient sampling noise as a bridge to precisely characterize each term. Let $\Delta_i^k := \nabla f_i(x^k)-\nabla f_i(w^k), \Delta^k := \nabla f(x^k)-\nabla f(w^k)$, and $\xi_{1i}^k := \Delta_i^k-\Delta^k, \xi_{2i}^k := \nabla f_i(w^k)-\nabla f(w^k)$, Then, we obtain:

$$
\begin{aligned}
\mathbb{E}_k[g^k] &= \mathbb{E}_k[\frac{1}{|S|}\sum_{i\in S}clip(\nabla f_i(x^k)-\nabla f_i(w^k), C_{1k}) + \frac{1}{|D|}\sum_{i\in D}clip(\nabla f_i(w^k), C_{2k})] \\
&= \mathbb{E}_k[\Delta_i^k \cdot min\{1, \frac{C_{1k}}{||\Delta_i^k||}\}] + \mathbb{E}_k[\nabla f_i(w^k) \cdot min\{1, \frac{C_{2k}}{||\nabla f_i(w^k)||}\}] \\
&= \mathbb{E}_k[(\Delta^k+\xi_{1i}^k) \cdot min\{1, \frac{C_{1k}}{||\Delta^k+\xi_{1i}^k||}\}] + \mathbb{E}_k[(\nabla f(w^k)+\xi_{2i}^k) \cdot min\{1, \frac{C_{2k}}{||\nabla f(w^k)+\xi_{2i}^k||}\}]
\end{aligned}
\tag{31}
$$

Therefore, for $\mathbb{E}_k\langle\nabla f(x^k), g^k\rangle$, we have:

$$
\begin{aligned}
\mathbb{E}_k\langle\nabla f(x^k), g^k\rangle = &\langle\nabla f(x^k), \mathbb{E}_k[(\Delta^k + \xi_{1i}^k) \cdot min\{1, \frac{C_{1k}}{||\Delta^k + \xi_{1i}^k||}\}] \\
&+ \mathbb{E}_k[(\nabla f(w^k) + \xi_{2i}^k) \cdot min\{1, \frac{C_{2k}}{||\nabla f(w^k) + \xi_{2i}^k||}\}]\rangle \\
= &\underbrace{\langle\nabla f(x^k), \mathbb{E}_k[(\Delta^k + \xi_{1i}^k) \cdot min\{1, \frac{C_{1k}}{||\Delta^k + \xi_{1i}^k||}\}] - \Delta^k\rangle}_{C:=Coupled\ Term} \\
&+ \underbrace{\langle\nabla f(x^k), \mathbb{E}_k[(\nabla f(w^k) + \xi_{2i}^k) \cdot min\{1, \frac{C_{2k}}{||\nabla f(w^k) + \xi_{2i}^k||}\}] - \nabla f(w^k) + \nabla f(x^k)\rangle}_{A:=Anchor\ Term}
\end{aligned}
\tag{32}
$$

We denote $\mathbb{P}_1^k := \mathbb{E}_k[1_{\{||\Delta^k + \xi_{1i}^k|| \le C_{1k}\}}]$, $\mathbb{P}_2^k := \mathbb{E}_k[1_{\{||\nabla f(w^k) + \xi_{2i}^k|| \le C_{2k}\}}]$, and assume that $C_1 > 1$ and $C_2 > 1$. We then examine the two terms separately. First, for the term **A**, we have:

$$
\begin{aligned}
\mathbb{E}_k[(\nabla f(w^k) + \xi_{2i}^k) \cdot min\{1, \frac{C_{2k}}{||\nabla f(w^k) + \xi_{2i}^k||}\}] = \\
\mathbb{E}_k[(\nabla f(w^k) + \xi_{2i}^k) \cdot 1_{\{||\nabla f(w^k) + \xi_{2i}^k|| \le C_{2k}\}}] + \mathbb{E}_k[\frac{C_{2k} \cdot (\nabla f(w^k) + \xi_{2i}^k)}{||\nabla f(w^k) + \xi_{2i}^k||} \cdot 1_{\{||\nabla f(w^k) + \xi_{2i}^k|| > C_{2k}\}}]
\end{aligned}
\tag{33}
$$

Substituting into (32), we obtain:

$$
\begin{aligned}
A = &||\nabla f(x^k)||^2 + \langle\nabla f(x^k), -\nabla f(w^k) + \mathbb{E}_k[(\nabla f(w^k) + \xi_{2i}^k) \cdot min\{1, \frac{C_{2k}}{||\nabla f(w^k) + \xi_{2i}^k||}\}]\rangle \\
= &||\nabla f(x^k)||^2 + \langle\nabla f(x^k), -\nabla f(w^k) + \mathbb{E}_k[(\nabla f(w^k) + \xi_i^k) \cdot 1_{\{||\nabla f(w^k) + \xi_{2i}^k|| \le C_{2k}\}}] \\
&+ \mathbb{E}_k[\frac{C_{2k} \cdot (\nabla f(w^k) + \xi_{2i}^k)}{||\nabla f(w^k) + \xi_{2i}^k||} \cdot 1_{\{||\nabla f(w^k) + \xi_{2i}^k|| > C_{2k}\}}]\rangle \\
= &||\nabla f(x^k)||^2 + \mathbb{E}_k[\langle\nabla f(x^k), -\nabla f(w^k) \cdot (1_{\{||\nabla f(w^k) + \xi_{2i}^k|| \le C_{2k}\}} + 1_{\{||\nabla f(w^k) + \xi_{2i}^k|| > C_{2k}\}}) \\
&+ (\nabla f(w^k) + \xi_{2i}^k) \cdot 1_{\{||\nabla f(w^k) + \xi_{2i}^k|| \le C_{2k}\}} + \frac{C_{2k} \cdot (\nabla f(w^k) + \xi_{2i}^k)}{||\nabla f(w^k) + \xi_{2i}^k||} \cdot 1_{\{||\nabla f(w^k) + \xi_{2i}^k|| > C_{2k}\}}\rangle] \\
= &||\nabla f(x^k)||^2 + \mathbb{E}_k[\langle\nabla f(x^k), \xi_{2i}^k \cdot 1_{\{||\nabla f(w^k) + \xi_{2i}^k|| \le C_{2k}\}}\rangle] \\
&+ \mathbb{E}_k[\langle\nabla f(x^k), [\frac{C_{2k} \cdot (\nabla f(w^k) + \xi_{2i}^k)}{||\nabla f(w^k) + \xi_{2i}^k||} - f(w^k)] \cdot 1_{\{||\nabla f(w^k) + \xi_{2i}^k|| > C_{2k}\}}\rangle]
\end{aligned}
\tag{34}
$$

Focusing on the final term of (34) alone, we obtain:

$$
\begin{aligned}
&\mathbb{E}_k[\langle\nabla f(x^k), [\frac{C_{2k} \cdot (\nabla f(w^k) + \xi_{2i}^k)}{||\nabla f(w^k) + \xi_{2i}^k||} - f(w^k)] \cdot 1_{\{||\nabla f(w^k) + \xi_{2i}^k|| > C_{2k}\}}\rangle] \\
= &\mathbb{E}_k[\langle\nabla f(x^k), [\frac{C_2||\nabla f(w^k)||(\nabla f(w^k) + \xi_{2i}^k)}{||\nabla f(w^k) + \xi_{2i}^k||} - (f(w^k) + \xi_{2i}^k) + \xi_{2i}^k] \cdot 1_{\{||\nabla f(w^k) + \xi_{2i}^k|| > C_{2k}\}}\rangle] \\
= &\mathbb{E}_k[\langle\nabla f(x^k), [\frac{[C_2||\nabla f(w^k)|| - ||\nabla f(w^k) + \xi_{2i}^k||](\nabla f(w^k) + \xi_{2i}^k)}{||\nabla f(w^k) + \xi_{2i}^k||} + \xi_{2i}^k] \cdot 1_{\{||\nabla f(w^k) + \xi_{2i}^k|| > C_{2k}\}}\rangle] \\
= &\mathbb{E}_k[\langle\nabla f(x^k), \frac{[C_2||\nabla f(w^k)|| - ||\nabla f(w^k) + \xi_{2i}^k||](\nabla f(w^k) + \xi_{2i}^k)}{||\nabla f(w^k) + \xi_{2i}^k||} \cdot 1_{\{||\nabla f(w^k) + \xi_{2i}^k|| > C_{2k}\}}\rangle] \\
&+ \mathbb{E}_k[\langle\nabla f(x^k), \xi_{2i}^k \cdot 1_{\{||\nabla f(w^k) + \xi_{2i}^k|| > C_{2k}\}}\rangle]
\end{aligned}
\tag{35}
$$

Substituting into (34), we obtain:

$$A = ||\nabla f(x^k)||^2 + \mathbb{E}_k[\langle \nabla f(x^k), \xi_{2i}^k \cdot (1_{\{||\nabla f(w^k) + \xi_{2i}^k|| \le C_{2k}\}} + 1_{\{||\nabla f(w^k) + \xi_{2i}^k|| > C_{2k}\}})\rangle]$$
$$+ \mathbb{E}_k[\langle \nabla f(x^k), \frac{[C_2||\nabla f(w^k)|| - ||\nabla f(w^k) + \xi_{2i}^k||](\nabla f(w^k) + \xi_{2i}^k)}{||\nabla f(w^k) + \xi_{2i}^k||} \cdot 1_{\{||\nabla f(w^k) + \xi_{2i}^k|| > C_{2k}\}}\rangle]$$
$$(36)$$

Since $1_{\{||\nabla f(w^k) + \xi_{2i}^k|| \le C_{2k}\}} + 1_{\{||\nabla f(w^k) + \xi_{2i}^k|| > C_{2k}\}} = 1$, and $C_2||\nabla f(w^k)|| - ||\nabla f(w^k) + \xi_{2i}^k|| \ge C_2||\nabla f(w^k)|| - ||\nabla f(w^k)|| - ||\xi_{2i}^k||$, together with the facts that $\mathbb{E}_k[\xi_{2i}^k] = 0$ and $(C_2||\nabla f(w^k)|| - ||\nabla f(w^k) + \xi_{2i}^k||) \cdot 1_{\{||\nabla f(w^k) + \xi_{2i}^k|| > C_{2k}\}} \le 0$, we can, by applying the Cauchy inequality, derive a lower bound for $A$:

$$A = ||\nabla f(x^k)||^2$$
$$+ \mathbb{E}_k[\underbrace{1_{\{||\nabla f(w^k) + \xi_{2i}^k|| > C_{2k}\}} \cdot [C_2||\nabla f(w^k)|| - ||\nabla f(w^k) + \xi_{2i}^k||]}_{<0} \cdot \langle \nabla f(x^k), \frac{(\nabla f(w^k) + \xi_{2i}^k)}{||\nabla f(w^k) + \xi_{2i}^k||}\rangle]$$
$$\overset{(a)}{\ge} ||\nabla f(x^k)||^2 + \mathbb{E}_k[1_{\{||\nabla f(w^k) + \xi_{2i}^k|| > C_{2k}\}} \cdot [(C_2-1)||\nabla f(w^k)|| - ||\xi_{2i}^k||] \cdot ||\nabla f(x^k)||]$$
$$\overset{(b)}{\ge} ||\nabla f(x^k)||^2 + \mathbb{E}_k[1_{\{||\nabla f(w^k) + \xi_{2i}^k|| > C_{2k}\}} \cdot [C_2||\nabla f(w^k)|| - ||\nabla f(w^k) + \xi_{2i}^k||] \cdot ||\nabla f(x^k)||]$$
$$= ||\nabla f(x^k)||^2 + \mathbb{E}_k[1_{\{||\nabla f(w^k) + \xi_{2i}^k|| > C_{2k}\}} \cdot (C_2-1)||\nabla f(w^k)|| \cdot ||\nabla f(x^k)||]$$
$$- \mathbb{E}_k[1_{\{||\nabla f(w^k) + \xi_{2i}^k|| > C_{2k}\}} \cdot ||\xi_{2i}^k|| \cdot ||\nabla f(x^k)||]$$
$$(37)$$

Here, $(a)$ follows from the Cauchy inequality, and $(b)$ follows from the triangle inequality. We now consider the third term in the above expression. Let $S_k$ denote the set of $\xi_{2i}^k$ such that $||\nabla f(w^k) + \xi_{2i}^k|| > C_{2k}$, and define $P_{k,z} := \Pr(\xi^k \in S_k, ||\xi^k|| = z)$. Then, we have:

$$- \mathbb{E}_k[1_{\{||\nabla f(w^k) + \xi_{2i}^k||_2 > C_{2k}\}} \cdot ||\xi_{2i}^k|| \cdot ||\nabla f(x^k)||]$$
$$= - \mathbb{E}_k[1_{\{\xi^k \in S_k\}} \cdot ||\xi^k|| \cdot ||\nabla f(x^k)||]$$
$$= - ||\nabla f(x^k)|| \cdot \int_0^{+\infty} P_{k,z} \cdot z \, dz$$
$$= - ||\nabla f(x^k)|| \cdot \int_0^{+\infty} \sqrt{P_{k,z}} \cdot \sqrt{z^2 \cdot P_{k,z}} \, dz \qquad (38)$$
$$\ge - ||\nabla f(x^k)|| \cdot \sqrt{(\int_0^{+\infty} P_{k,z} \, dz) \cdot (\int_0^{+\infty} z^2 P_{k,z} \, dz)}$$
$$\ge - ||\nabla f(x^k)|| \cdot \sqrt{(1 - \mathbb{P}_2^k)} \cdot \sqrt{\mathbb{E}_k[||\xi_{2i}^k||^2]}$$

Thus, we can obtain:

$$A \ge ||\nabla f(x^k)||^2 + (1 - \mathbb{P}_2^k)(C_2-1)||\nabla f(x^k)|| \cdot ||\nabla f(w^k)|| - ||\nabla f(x^k)|| \cdot \sqrt{(1 - \mathbb{P}_2^k)} \cdot \sqrt{\mathbb{E}_k[||\xi_{2i}^k||^2]}$$
$$(39)$$

The treatment of $C$ is analogous to that of $A$. Due to the symmetry between $\nabla f(w^k)$ and $\Delta^k$ in the expressions for $A$ and $C$, and by referring to (32)–(39), we can obtain:

$$
\begin{aligned}
C &= \langle \nabla f(x^k), \mathbb{E}_k[\frac{\Delta^k + \xi_{1i}^k}{||\Delta^k + \xi_{1i}^k||} \cdot \underbrace{(C_{1k} - ||\Delta^k + \xi_{1i}^k||) \cdot 1_{\{||\Delta^k + \xi_{1i}^k|| > C_{1k}\}}}_{<0}]\rangle \\
&\geq \mathbb{E}_k[(C_{1k} - ||\Delta^k + \xi_{1i}^k||) \cdot 1_{\{||\Delta^k + \xi_{1i}^k|| > C_{1k}\}}] \cdot ||\nabla f(x^k)|| \\
&\geq \mathbb{E}_k[(C_1||\Delta^k|| - ||\Delta^k|| - ||\xi_{1i}^k||) \cdot 1_{\{||\Delta^k + \xi_{1i}^k|| > C_{1k}\}}] \cdot ||\nabla f(x^k)|| \\
&\geq \mathbb{E}_k[(C_1 - 1)||\Delta^k|| \cdot 1_{\{||\Delta^k + \xi_{1i}^k|| > C_{1k}\}}] \cdot ||\nabla f(x^k)|| \\
&\quad - \mathbb{E}_k[||\nabla f(x^k)|| \cdot ||\xi_{1i}^k|| \cdot 1_{\{||\Delta^k + \xi_{1i}^k|| > C_{1k}\}}] \\
&\geq (1 - \mathbb{P}_1^k)(C_1 - 1)||\Delta^k||||\nabla f(x^k)|| - ||\nabla f(x^k)|| \cdot \sqrt{(1 - \mathbb{P}_1^k)} \cdot \sqrt{\mathbb{E}_k[||\xi_{1i}^k||^2]} \\
&\geq (1 - \mathbb{P}_1^k)(C_1 - 1)||\nabla f(x^k) - \nabla f(w^k)|| \cdot ||\nabla f(x^k)|| \\
&\quad - ||\nabla f(x^k)|| \cdot \sqrt{(1 - \mathbb{P}_1^k)} \cdot \sqrt{\mathbb{E}_k[||\xi_{1i}^k||^2]}
\end{aligned} \tag{40}
$$

By Assumption 3.4, we have $\mathbb{E}_k[||\xi_{2i}^k||^2] = \mathrm{Var}(\nabla f_i) \leq \tau^2$. Moreover, $\xi_{1i}^k = \Delta_i^k - \Delta^k = \nabla f_i(x^k) - \nabla f_i(w^k) - \nabla f(x^k) + \nabla f(w^k) = (\nabla f_i(x^k) - \nabla f(x^k)) - (\nabla f_i(w^k) - \nabla f(w^k)) = \xi_{2i}^k - \xi_{2i}^{k'}$, which implies $\mathbb{E}_k||\xi_{1i}^k||^2 \leq 4\tau^2$. Combining the results from both terms, we obtain:

$$
\begin{aligned}
\mathbb{E}_k[\langle \nabla f(x^k), g^k \rangle] &\geq ||\nabla f(x^k)||^2 + (1 - \mathbb{P}_1^k)(C_1 - 1)||\nabla f(x^k)|| \cdot ||\nabla f(x^k) - \nabla f(w^k)|| \\
&\quad + (1 - \mathbb{P}_2^k)(C_2 - 1)||\nabla f(x^k)|| \cdot ||\nabla f(w^k)|| - 2||\nabla f(x^k)|| \cdot \sqrt{(1 - \mathbb{P}_1^k)} \cdot \tau \\
&\quad - ||\nabla f(x^k)|| \cdot \sqrt{(1 - \mathbb{P}_2^k)} \cdot \tau
\end{aligned} \tag{41}
$$

Below, we consider two cases, namely $||\nabla f(x^k)|| \geq 3\tau$ and $||\nabla f(x^k)|| < 3\tau$, and we will use probabilities to combine them. For the former case, we have:

$$
\begin{aligned}
\mathbb{E}_k[\langle \nabla f(x^k), g^k \rangle] &\geq (1 - \frac{2\sqrt{1 - \mathbb{P}_1^k} + \sqrt{1 - \mathbb{P}_2^k}}{3})||\nabla f(x^k)||^2 \\
&\quad + \frac{2\sqrt{1 - \mathbb{P}_1^k} + \sqrt{1 - \mathbb{P}_2^k}}{3}||\nabla f(x^k)||^2 \\
&\quad - (2\sqrt{(1 - \mathbb{P}_2^k)} + \sqrt{(1 - \mathbb{P}_1^k)})||\nabla f(x^k)||\tau \\
&\quad + (1 - \mathbb{P}_2^k)(C_2 - 1)||\nabla f(x^k)|| \cdot ||\nabla f(w^k)|| \\
&\quad + (1 - \mathbb{P}_1^k)(C_1 - 1)||\nabla f(x^k)|| \cdot ||\nabla f(x^k) - \nabla f(w^k)|| \\
&\geq (1 - \frac{2\sqrt{1 - \mathbb{P}_1^k} + \sqrt{1 - \mathbb{P}_2^k}}{3})||\nabla f(x^k)||^2 \\
&\quad + (1 - \mathbb{P}_2^k)(C_2 - 1)||\nabla f(x^k)|| \cdot ||\nabla f(w^k)|| \\
&\quad + (1 - \mathbb{P}_1^k)(C_1 - 1)||\nabla f(x^k)|| \cdot ||\nabla f(x^k) - \nabla f(w^k)|| \geq 0
\end{aligned} \tag{42}
$$

In summary, by combining the two cases using probabilities, let $\mathbb{P}^k := \Pr(\|\nabla f(x^k)\| < 3\tau \mid x^{k-1})$. Then, we have:

$$
\begin{aligned}
\mathbb{E}_k[\langle \nabla f(x^k), g^k\rangle] &\geq (1 - \mathbb{P}^k)(1 - \frac{2\sqrt{1 - \mathbb{P}_1^k} + \sqrt{1 - \mathbb{P}_2^k}}{3})\|\nabla f(x^k)\|^2 \\
&\quad + (1 - \mathbb{P}^k)(1 - \mathbb{P}_2^k)(C_2 - 1)\|\nabla f(x^k)\| \cdot \|\nabla f(w^k)\| \\
&\quad + (1 - \mathbb{P}^k)(1 - \mathbb{P}_1^k)(C_1 - 1)\|\nabla f(x^k)\| \cdot \|\nabla f(x^k) - \nabla f(w^k)\| \\
&\quad + \mathbb{P}^k\|\nabla f(x^k)\|^2 + \mathbb{P}^k(1 - \mathbb{P}_1^k)(C_1 - 1)\|\nabla f(x^k)\| \cdot \|\nabla f(x^k) - \nabla f(w^k)\| \\
&\quad + \mathbb{P}^k(1 - \mathbb{P}_2^k)(C_2 - 1)\|\nabla f(x^k)\| \cdot \|\nabla f(w^k)\| - 2\mathbb{P}^k\|\nabla f(x^k)\| \cdot \sqrt{(1 - \mathbb{P}_1^k)} \cdot \tau \\
&\quad - \mathbb{P}^k\|\nabla f(x^k)\| \cdot \sqrt{(1 - \mathbb{P}_2^k)} \cdot \tau \\
&= \underbrace{(1 - (1 - \mathbb{P}^k)\frac{2\sqrt{1 - \mathbb{P}_1^k} + \sqrt{1 - \mathbb{P}_2^k}}{3})}_{0 \leq \lambda_1^k \leq 1}\|\nabla f(x^k)\|^2 \\
&\quad + \underbrace{(1 - \mathbb{P}_2^k)(C_2 - 1)}_{\lambda_2^k \geq 0}\|\nabla f(x^k)\| \cdot \|\nabla f(w^k)\| \\
&\quad + \underbrace{(1 - \mathbb{P}_1^k)(C_1 - 1)}_{\lambda_3^k \geq 0}\|\nabla f(x^k)\|_2 \cdot \|\nabla f(x^k) - \nabla f(w^k)\| \\
&\quad - \underbrace{\mathbb{P}^k \cdot \frac{2\sqrt{1 - \mathbb{P}_1^k} + \sqrt{1 - \mathbb{P}_2^k}}{3}}_{0 \leq \lambda_4^k \leq 1} \cdot \|\nabla f(x^k)\| \cdot 3\tau
\end{aligned}
\tag{43}
$$

For notational convenience, we further define and restate:

$$
\lambda_1^k := 1 - \frac{1}{3}(1 - \mathbb{P}^k)(2\sqrt{1 - \mathbb{P}_1^k} + \sqrt{1 - \mathbb{P}_2^k}), \quad \lambda_2^k := (1 - \mathbb{P}_2^k)(C_2 - 1), \quad \lambda_3^k := (1 - \mathbb{P}_1^k)(C_1 - 1)
$$

$$
\lambda_4^k := \frac{1}{3}\mathbb{P}^k(2\sqrt{1 - \mathbb{P}_1^k} + \sqrt{1 - \mathbb{P}_2^k}), \quad \mathbb{P}^k := \Pr\left(\|\nabla f(x^k)\| \leq 3\tau \mid x^{k-1}\right),
$$

$$
\mathbb{P}_1^k := \mathbb{E}_k\left[\mathbb{1}_{\{\|\nabla f_i(x^k) - \nabla f_i(w^k)\|_2 \leq C_{1k}\}}\right], \quad \mathbb{P}_2^k := \mathbb{E}_k\left[\mathbb{1}_{\{\|\nabla f_i(w^k)\|_2 \leq C_{2k}\}}\right].
$$

By substituting (43), (30) into (29), taking the full expectation on both sides of the inequality, summing over $k = 1$ to $T$, and setting $\eta = \sqrt{\frac{2(f(x^0) - f(x^\star))}{TL\tilde{G}}}$, we obtain:

$$
\begin{aligned}
\frac{1}{T}\sum_{k=1}^{T}\mathbb{E}&\left[\lambda_1^k\|\nabla f(x^k)\|^2 + \lambda_2^k\|\nabla f(x^k)\| \cdot \|\nabla f(w^k)\| + \lambda_3^k\|\nabla f(x^k)\| \cdot \|\nabla f(x^k) - \nabla f(w^k)\|\right] \\
&\leq \frac{f(x^0) - \mathbb{E}f(x^T)}{\eta T} + \frac{\eta L}{2T}\sum_{k=1}^{T}\tilde{G} + \frac{1}{T}\sum_{k=1}^{T}\mathbb{E}\left[\lambda_4^k \cdot 3\tau\|\nabla f(x^k)\|\right] \\
&\leq 2\sqrt{\frac{(f(x^0) - f(x^\star))L\tilde{G}}{2T}} + \frac{1}{T}\sum_{k=1}^{T}\mathbb{E}\left[\lambda_4^k \cdot 3\tau\|\nabla f(x^k)\|\right]
\end{aligned}
\tag{44}
$$

### C.4 PROOF OF THEOREM 4

The proof of Thm.4 is similar to that of Thm.1. Following the previous approach, we focus mainly on presenting the differences. The treatment of expectations is similar; for simplicity, we do not distinguish them in the notation. Continuing from (14), we first consider the upper bound of $\mathbb{E}\|\tilde{g}^k\|^2$.

For any precision $e > 0$, when $\|\nabla f(w)\|, \|x^k - x^\star\|, \|w^k - x^\star\| > e$, we have:

$$
\begin{aligned}
\mathbb{E}\|\tilde{g}^k\|^2 &= \mathbb{E}[\|\frac{1}{|S|}\sum_{i \in S}clip(\nabla f_i(x^k) - \nabla f_i(w^k), C_1\|x^k - w^k\|) \\
&\quad + \frac{1}{N}\sum_{i \in D}clip(\nabla f_i(w^k), C_2\|\nabla f(w^{k-1})\|) + \mathrm{n}_1^k + \mathrm{n}_2^k\|^2] \\
&\leq 4\mathbb{E}\|\frac{1}{|S|}\sum_{i \in S}clip(\nabla f_i(x^k) - \nabla f_i(w^k), C_1\|x^k - w^k\|)\|^2 \\
&\quad + 4\mathbb{E}\|\frac{1}{|D|}\sum_{i \in D}clip(\nabla f_i(w^k), C_2\|\nabla f(w^{k-1})\|)\|^2 \\
&\quad + 4d\sigma_1^2 C_1^2\mathbb{E}\|x^k - w^k\|^2 + 4d\sigma_2^2 C_2^2\mathbb{E}\|\nabla f(w^{k-1})\|^2 \\
&\leq 4C_1^2(d\sigma_1^2 + 1)\mathbb{E}\|x^k - x^\star + x^\star - w^k\|^2 \\
&\quad + 4C_2^2(d\sigma_2^2 + 1)\mathbb{E}\|\nabla f(w^k) - (\nabla f(w^k) - \nabla f(w^{k-1}))\|^2 \\
&\leq 8C_1^2(d\sigma_1^2 + 1)\mathbb{E}\|x^k - x^\star\|^2 + 8C_1^2(d\sigma_1^2 + 1)\mathbb{E}\|w^k - x^\star\|^2 \\
&\quad + 8C_2^2(d\sigma_2^2 + 1)\mathbb{E}\|\nabla f(w^k)\|^2 + 8C_2^2(d\sigma_2^2 + 1)\mathbb{E}\|\nabla f(w^k) - \nabla f(w^{k-1})\|^2 \\
&\overset{(a)}{\leq} 8C_1^2(d\sigma_1^2 + 1)\mathbb{E}\|x^k - x^\star\|^2 + 8C_1^2(d\sigma_1^2 + 1)\mathbb{E}\|w^k - x^\star\|^2 \\
&\quad + 8C_2^2(d\sigma_2^2 + 1)\mathbb{E}\|\nabla f(w^k)\|^2 + 8C_2^2 L^2(d\sigma_2^2 + 1)\mathbb{E}\|w^k - w^{k-1}\|^2 \\
&\overset{(b)}{\leq} 8C_1^2(d\sigma_1^2 + 1)\mathbb{E}\|x^k - x^\star\|^2 + 8C_1^2(d\sigma_1^2 + 1)\mathbb{E}\|w^k - x^\star\|^2 \\
&\quad + 8C_2^2(d\sigma_2^2 + 1)\mathbb{E}\|\nabla f(w^k)\|^2 + 8C_2^2 L^2(d\sigma_2^2 + 1)\eta^2 \frac{\tilde{G}^2}{pe^2} \cdot e^2 \\
&\leq 8C_1^2(d\sigma_1^2 + 1)\mathbb{E}\|x^k - x^\star\|^2 + 8C_1^2(d\sigma_1^2 + 1)\mathbb{E}\|w^k - x^\star\|^2 \\
&\quad + 8C_2^2(d\sigma_2^2 + 1)\mathbb{E}\|\nabla f(w^k)\|^2 + 4C_2^2 L^2(d\sigma_2^2 + 1)\eta^2 \frac{\tilde{G}^2}{pe^2}(\mathbb{E}\|x^k - x^\star\|^2 + \mathbb{E}\|w^k - x^\star\|^2) \\
&\leq \underbrace{(8C_1^2(d\sigma_1^2 + 1) + \frac{4\eta^2}{pe^2}G^2 C_2^2(d\sigma_2^2 + 1))}_{:= N_1}(\mathbb{E}\|x^k - x^\star\|^2 + \mathbb{E}\|w^k - x^\star\|^2) \\
&\quad + \underbrace{8C_2^2(d\sigma_2^2 + 1)}_{:= N_2}\underbrace{\mathbb{E}\|\nabla f_i(w^k) - \nabla f_i(x^\star)\|^2}_{:= D^k}.
\end{aligned}
\tag{45}
$$

Where, (a) follows from the $L$-smooth property, and (b) follows from (22), $\tilde{G}$ is given by (46). Similarly, for $\mathbb{E}\|\tilde{g}^k\|$, we have:

$$
\begin{aligned}
\mathbb{E}\|\tilde{g}^k\| &\leq \sqrt{\mathbb{E}\|\tilde{g}^k\|^2} \\
&\leq (4C_1^2(d\sigma_1^2 + 1)\|x^k - w^k\|^2 + 4C_2^2(d\sigma_2^2 + 1)\|\nabla f(w^{k-1})\|^2)^{\frac{1}{2}} \\
&\leq (\frac{4C_1^2(d\sigma_1^2 + 1)}{\mu^2}\|\nabla f(x^k) - \nabla f(w^k)\|^2 + 4C_2^2(d\sigma_2^2 + 1)\|\nabla f(w^{k-1})\|^2)^{\frac{1}{2}} \\
&\leq (\frac{8C_1^2(d\sigma_1^2 + 1)G^2}{\mu^2} + 4C_2^2(d\sigma_2^2 + 1)G^2)^{\frac{1}{2}} \\
&= \frac{2G}{\mu}\sqrt{2C_1^2(d\sigma_1^2 + 1) + \mu^2 C_2^2(d\sigma_2^2 + 1)} := \tilde{G}
\end{aligned}
\tag{46}
$$

Similarly, for the two types of clipping bias, we have:

$$
b_1^k := \frac{1}{|S|}\sum_{i \in S}clip(\Delta_i^k, C_1\|x^k - w^k\|) - \Delta_S^k,
$$

$$
b_2^k := \frac{1}{|D|}\sum_{i \in D}clip(\nabla f_i(w^k), C_2\|\nabla f(w^{k-1})\|) - \nabla f(w^k)
$$

Following all the previously introduced notations, we have:

$$b_1^k = \frac{1}{|S|} \sum_{i \in S} clip(\Delta_i^k, C_1||x^k - w^k||) - \Delta_S^k$$

$$= \frac{1}{|S|} \left( \sum_{i \in J_1^k} \Delta_i^k + \sum_{i \in J_2^k} \frac{C_{1k}}{||\Delta_i^k||} \Delta_i^k \right) - \Delta_S^k$$

$$= \frac{1}{|S|} \sum_{i \in J_2^k} \left( \frac{C_{1k}}{||\Delta_i^k||} - 1 \right) \cdot \Delta_i^k \qquad (47)$$

$$= \frac{1}{|S|} \sum_{i \in J_2^k} \underbrace{(C_1||x^k - w^k|| - ||\Delta_i^k||)}_{<0} \cdot \frac{\Delta_i^k}{||\Delta_i^k||}$$

$$b_2^k = \frac{1}{|D|} \sum_{i \in J_2^k} \underbrace{(C_2||\nabla f(w^{k-1})|| - ||\nabla f_i(w^k)||)}_{<0} \cdot \frac{\nabla f_i(w^k)}{||\nabla f_i(w^k)||}$$

Similarly, for the first type of clipping, we define $\mathbb{P}_1^k := \mathbb{E}_k[1_{\{||\Delta_i^k|| \le C_{1k}\}}]$, and we have:

$$\mathbb{E}[\langle b_1^k, x^\star - x^k \rangle] = \mathbb{E}[\langle \frac{1}{|S|} \sum_{i \in J_2^k} \underbrace{(C_1||x^k - w^k|| - ||\Delta_i^k||)}_{<0} \cdot \frac{\Delta_i^k}{||\Delta_i^k||}, x^\star - x^k \rangle]$$

$$\le \mathbb{E}[\frac{1}{|S|} \sum_{i \in J_2^k} (||\Delta_i^k|| - C_1||x^k - w^k||) \cdot ||x^k - x^\star||]$$

$$\le \mathbb{E}[\frac{1}{|S|} \sum_{i \in J_2^k} (L||x^k - w^k|| - C_1||x^k - w^k||) \cdot ||x^k - x^\star||] \qquad (48)$$

$$\le \mathbb{E}(1 - \mathbb{P}_1^k)(L - C_1)||x^k - w^k|| \cdot ||x^k - x^\star||$$

$$\overset{(a)}{\le} \eta(L - C_1) \frac{G\tilde{G}}{p\mu e^2} \mathbb{E}||x^k - x^\star||^2$$

Here, (a) follows directly from (21). For the second type of clipping, we define $\mathbb{P}_2^k := \mathbb{E}_k[1_{\{||\nabla f_i(w^k)|| \le C_{2k}\}}]$, and we have:

$$\mathbb{E}\langle b_2^k, x^\star - x^k \rangle = \mathbb{E}\langle \frac{1}{|D|} \sum_{i \in J_2^k} \underbrace{(C_2||\nabla f(w^{k-1})|| - ||\nabla f_i(w^k)||)}_{<0} \cdot \frac{\nabla f_i(w^k)}{||\nabla f_i(w^k)||}, x^\star - x^k \rangle$$

$$\le \mathbb{E} \frac{1}{|D|} \sum_{i \in J_2^k} (||\nabla f_i(w^k)|| - C_2||\nabla f(w^{k-1})||) \cdot ||x^k - x^\star||]$$

$$\le \mathbb{E} \frac{1}{|D|} \sum_{i \in J_2^k} (||\nabla f_i(w^k)|| - C_2||\nabla f(w^k)|| + C_2||\nabla f(w^k) - \nabla f(w^{k-1})||) \cdot ||x^k - x^\star||$$

$$\le \mathbb{E} \frac{1}{|D|} \sum_{i \in J_2^k} (\underbrace{(\tau - (C_2 - 1)||\nabla f(w^k)||)}_{\le 0} + C_2||\nabla f(w^k) - \nabla f(w^{k-1})||) \cdot ||x^k - x^\star||$$

$$\le \mathbb{E} \frac{1}{|D|} \sum_{i \in J_2^k} C_2||\nabla f(w^k) - \nabla f(w^{k-1})|| \cdot ||x^k - x^\star||$$

$$\le \mathbb{E}(1 - \mathbb{P}_2^k) L C_2 ||w^k - w^{k-1}|| \cdot ||x^k - x^\star||$$

$$\overset{(a)}{\le} L p C_2 \mathbb{E}||x^{k-1} - w^{k-1}|| \cdot ||x^k - x^\star||$$

$$\overset{(b)}{\le} \eta L p C_2 \frac{G\tilde{G}}{p\mu e^2} \cdot e^2$$

$$\overset{(c)}{\le} \eta L C_2 \frac{G\tilde{G}}{\mu e^2} \cdot \mathbb{E}||x^k - x^\star||^2$$

$$(49)$$

Here, (a) comes from the iterative update rule of $w^k$; (b) follows directly from (22); and (c) is due to the precision conditions we imposed. In summary, let $A := \frac{2G\tilde{G}}{p\mu e^2}[(pC_2+1)L-C_1]$. We then consider the worst case, i.e., $A > 0$. In this case, similarly, we have:

$$
\begin{aligned}
\mathbb{E}||x^{k+1}-x^*||^2 &= \mathbb{E}||x^k-x^*-\eta\tilde{g}^k||^2 \\
&= \mathbb{E}||x^k-x^*||^2 + \mathbb{E}[2\eta\langle\tilde{g}^k, x^*-x^k\rangle] + \eta^2\mathbb{E}||\tilde{g}^k||^2 \\
&\leq \mathbb{E}||x^k-x^*||^2 + 2\eta\mathbb{E}\langle\nabla f(x^k)+b_1^k+b_2^k, x^*-x^k\rangle + \eta^2\mathbb{E}||\tilde{g}^k||^2 \\
&\leq \mathbb{E}||x^k-x^*||^2 + 2\eta\underbrace{(f^*-\mathbb{E}f(x^k)-(\frac{\mu}{2}-\eta\frac{A}{2})\mathbb{E}||x^k-x^*||^2)}_{\mu-strongly\ convex} + \eta^2\mathbb{E}||\tilde{g}^k||^2 \\
&= \mathbb{E}||x^k-x^*||^2(1-\eta\mu+\eta^2A) + 2\eta(f^*-\mathbb{E}f(x^k)) + \eta^2\mathbb{E}||\tilde{g}^k||^2
\end{aligned}
\tag{50}
$$

In (45), let $D^k := \mathbb{E}||\nabla f_i(w^k) - \nabla f_i(x^\star)||^2$, $N_1 := 8C_1^2(d\sigma_1^2+1) + \frac{4\eta^2}{pe^2}G^2C_2^2(d\sigma_2^2+1)$, $N_2 := 8C_2^2(d\sigma_2^2+1)$. Substituting these into (50), we obtain:

$$
\begin{aligned}
\mathbb{E}||x^{k+1}-x^\star||^2 &\leq (1-\eta\mu+\eta^2(N_1+A))\mathbb{E}||x^k-x^\star||^2 + \eta^2N_1\mathbb{E}||w^k-x^\star||^2 \\
&\quad + \eta^2N_2D^k - 2\eta(\mathbb{E}f(x^k)-f^\star)
\end{aligned}
\tag{51}
$$

Similarly, from the iterative update rule, we have:

$$
\begin{aligned}
\mathbb{E}||w^{k+1}-x^\star||^2 &= p\mathbb{E}||x^k-x^\star||^2 + (1-p)\mathbb{E}||w^k-x^\star||^2 \\
D^{k+1} &= (1-p)D^k + p\mathbb{E}||\nabla f_i(x^k)-\nabla f_i(x^\star)||^2 \\
&\leq (1-p)D^k + 2Lp(\mathbb{E}f(x^k)-f^\star)
\end{aligned}
\tag{52}
$$

We define the Lyapunov function of the system as follows:

$$
\Phi^k = \mathbb{E}||x^k-x^\star||^2 + \frac{2N_1\eta^2}{p}\mathbb{E}||w^k-x^\star||^2 + \frac{2N_2\eta^2}{p}D^k
\tag{53}
$$

Similarly, let $\eta < \min\{\frac{\mu}{3N_1+A}, \frac{1}{2LN_2}\}$, then we have:

$$
\begin{aligned}
\Phi^{k+1} &= \mathbb{E}||x^{k+1}-x^\star||^2 + \frac{2N_1\eta^2}{p}\mathbb{E}||w^{k+1}-x^\star||^2 + \frac{2N_2\eta^2}{p}D^{k+1} \\
&\leq (1-\mu\eta+(N_1+A)\eta^2+p\frac{2N_1\eta^2}{p})\mathbb{E}||x^k-x^\star||^2 + (N_1\eta^2+(1-p)\frac{2N_1\eta^2}{p})\mathbb{E}||w^k-x^\star||^2 \\
&\quad + (N_2\eta^2+(1-p)\frac{2N_2\eta^2}{p})D^k + \underbrace{(4LN_2\eta^2-2\eta)}_{<0}(\mathbb{E}f(x^k)-f^\star) \\
&= \underbrace{(1-\mu\eta+(3N_1+A)\eta^2)}_{<1}\mathbb{E}||x^k-x^\star||^2 + (1-\frac{p}{2})\frac{2N_1\eta^2}{p}\mathbb{E}||w^k-x^\star||^2 + (1-\frac{p}{2})\frac{2N_2\eta^2}{p}D^k
\end{aligned}
\tag{54}
$$

From this we can obtain the following:

$$
\Phi^{k+1} \leq max\{1-\mu\eta+(3N_1+A)\eta^2, 1-\frac{p}{2}\} \cdot \Phi^k < \Phi^k
\tag{55}
$$

which implies an exponential decay of the Lyapunov function.

## C.5 PROOF OF THEOREM 5

Similar to DP-C4, we first derive the upper bound of $\mathbb{E}||\tilde{g}^k||^2$. From (45), we have:

$$
\begin{aligned}
\mathbb{E}||\tilde{g}^k||^2 &\leq 4C_1^2(d\sigma_1^2+1)||x^k-w^k||^2 + 4C_2^2(d\sigma_2^2+1)||\nabla f(w^{k-1})||^2 \\
&\leq 4C^2(d\sigma_1^2+1) + 4G^2C_2^2(d\sigma_2^2+1) := \tilde{G}
\end{aligned}
\tag{56}
$$

Next, we discuss the lower bound of $\mathbb{E}\langle \nabla f(x^k), g^k \rangle$. Let $\Delta_i^k := \nabla f_i(x^k) - \nabla f_i(w^k), \Delta^k := \nabla f(x^k) - \nabla f(w^k), \xi_{1i}^k := \Delta_i^k - \Delta^k, \xi_{2i}^k := \nabla f_i(w^k) - \nabla f(w^k)$. Similarly, we can obtain:

$$\mathbb{E}\langle \nabla f(x^k), g^k \rangle$$

$$= \mathbb{E}\langle \nabla f(x^k), \mathbb{E}_k[(\Delta^k + \xi_{1i}^k) \cdot min\{1, \frac{C_{1k}}{||\Delta^k + \xi_{1i}^k||}\} + (\nabla f(w^k) + \xi_{2i}^k) \cdot min\{1, \frac{C_{2k}}{||\nabla f(w^k) + \xi_{2i}^k||}\}]\rangle$$

$$= \underbrace{\mathbb{E}\langle \nabla f(x^k), \mathbb{E}_k[(\Delta^k + \xi_{1i}^k) \cdot min\{1, \frac{C_{1k}}{||\Delta^k + \xi_{1i}^k||}\}] - \Delta^k\rangle}_{C := Coupled\ Term}$$

$$+ \underbrace{\mathbb{E}\langle \nabla f(x^k), \mathbb{E}_k[(\nabla f(w^k) + \xi_{2i}^k) \cdot min\{1, \frac{C_{2k}}{||\nabla f(w^k) + \xi_{2i}^k||}\}] - \nabla f(w^k) + \nabla f(x^k)\rangle}_{A := Anchor\ Term}$$

$$(57)$$

We denote $\mathbb{P}_1^k := \mathbb{E}_k[1_{\{||\Delta^k + \xi_{1i}^k|| \le C_{1k}\}}], \mathbb{P}_2^k := \mathbb{E}_k[1_{\{||\nabla f(w^k) + \xi_{2i}^k|| \le C_{2k}\}}]$, and assume that $C_1 > 1$ and $C_2 > 1$. Similarly, for the Anchor Term $A$, we have:

$$A = \mathbb{E}||\nabla f(x^k)||^2 + \mathbb{E}\Big[\mathbb{E}_k \underbrace{1_{\{||\nabla f(w^k) + \xi_{2i}^k|| > C_{2k}\}}(C_2||\nabla f(w^{k-1})|| - ||\nabla f(w^k) + \xi_{2i}^k||)\langle \nabla f(x^k), \frac{\nabla f(w^k) + \xi_{2i}^k}{||\nabla f(w^k) + \xi_{2i}^k||}\rangle}_{<0}\Big]$$

$$\ge \mathbb{E}||\nabla f(x^k)||^2 + \mathbb{E}\Big[\mathbb{E}_k 1_{\{||\nabla f(w^k) + \xi_{2i}^k|| > C_{2k}\}} \cdot (C_2||\nabla f(w^{k-1})|| - ||\nabla f(w^k) + \xi_{2i}^k||) \cdot ||\nabla f(x^k)||\Big]$$

$$\ge \mathbb{E}||\nabla f(x^k)||^2 + \mathbb{E}\Big[\mathbb{E}_k 1_{\{||\nabla f(w^k) + \xi_{2i}^k|| > C_{2k}\}} \cdot (C_2||\nabla f(w^{k-1})|| - ||\nabla f(w^k)|| - ||\xi_{2i}^k||) \cdot ||\nabla f(x^k)||\Big]$$

$$\ge \mathbb{E}||\nabla f(x^k)||^2 + \mathbb{E}\Big[\mathbb{E}_k 1_{\{||\nabla f(w^k) + \xi_{2i}^k|| > C_{2k}\}} \cdot (C_2||\nabla f(w^k)||$$

$$- C_2||\nabla f(w^k) - \nabla f(w^{k-1})|| - ||\nabla f(w^k)|| - ||\xi_{2i}^k||) \cdot ||\nabla f(x^k)||\Big]$$

$$\overset{(a)}{\ge} \mathbb{E}||\nabla f(x^k)||^2 + \underbrace{\mathbb{E}\Big[\mathbb{E}_k 1_{\{||\nabla f(w^k) + \xi_{2i}^k|| > C_{2k}\}} \cdot ((C_2 - 1)||\nabla f(w^k)|| - ||\xi_{2i}^k||) \cdot ||\nabla f(x^k)||\Big]}_{same\ as\ DP-C4}$$

$$- \mathbb{E}\Big[\mathbb{E}_k 1_{\{||\nabla f(w^k) + \xi_{2i}^k|| > C_{2k}\}} L||w^k - w^{k-1}|| \cdot ||\nabla f(x^k)||\Big]$$

$$\ge \mathbb{E}||\nabla f(x^k)||^2 + \mathbb{E}\Big[(1 - \mathbb{P}_2^k)(C_2 - 1)||\nabla f(x^k)|| \cdot ||\nabla f(w^k)||\Big] - \mathbb{E}\Big[||\nabla f(x^k)|| \cdot \tau\sqrt{1 - \mathbb{P}_2^k}\Big]$$

$$- Lp\mathbb{E}\Big[||x^{k-1} - w^{k-1}|| \cdot ||\nabla f(x^k)||\Big]$$

$$\overset{(b)}{\ge} \mathbb{E}||\nabla f(x^k)||^2 + \mathbb{E}\Big[(1 - \mathbb{P}_2^k)(C_2 - 1)||\nabla f(x^k)|| \cdot ||\nabla f(w^k)||\Big] - \mathbb{E}\Big[||\nabla f(x^k)|| \cdot \tau\sqrt{1 - \mathbb{P}_2^k}\Big] - \eta \cdot LG\tilde{G}$$

$$(58)$$

Here, (a) follows the same treatment as in DP-C4, and (b) can be directly obtained from (22). Similarly, for the Coupled Term $C$, we have:

$$C = \mathbb{E}\langle \nabla f(x^k), \mathbb{E}_k\Big[\frac{\Delta^k + \xi_{1i}^k}{||\Delta^k + \xi_{1i}^k||} \cdot \underbrace{(C_{1k} - ||\Delta^k + \xi_{1i}^k||) \cdot 1_{\{||\Delta^k + \xi_{1i}^k|| > C_{1k}\}}}_{<0}\Big]\rangle$$

$$\ge \mathbb{E}\Big[\mathbb{E}_k[(C_{1k} - ||\Delta^k + \xi_{1i}^k||) \cdot 1_{\{||\Delta^k + \xi_{1i}^k|| > C_{1k}\}}] \cdot ||\nabla f(x^k)||\Big]$$

$$\ge \mathbb{E}\Big[\mathbb{E}_k[(C_1||x^k - w^k|| - ||\Delta^k|| - ||\xi_{1i}^k||) \cdot 1_{\{||\Delta^k + \xi_{1i}^k|| > C_{1k}\}}] \cdot ||\nabla f(x^k)||\Big]$$

$$\ge \mathbb{E}\Big[(\frac{C_1}{L} - 1)||\Delta^k|| \cdot \mathbb{E}_k 1_{\{||\Delta^k + \xi_{1i}^k|| > C_{1k}\}}] \cdot ||\nabla f(x^k)||\Big] - \mathbb{E}\Big[||\nabla f(x^k)|| \cdot \mathbb{E}_k\big[||\xi_{1i}^k|| \cdot 1_{\{||\Delta^k + \xi_{1i}^k|| > C_{1k}\}}\big]\Big]$$

$$\ge \mathbb{E}(1 - \mathbb{P}_1^k)(\frac{C_1}{L} - 1)||\Delta^k|| \cdot ||\nabla f(x^k)|| - \mathbb{E}||\nabla f(x^k)|| \cdot \sqrt{(1 - \mathbb{P}_1^k)} \cdot \sqrt{\mathbb{E}_k[||\xi_{1i}^k||^2]}$$

$$\ge \mathbb{E}(1 - \mathbb{P}_1^k)(\frac{C_1}{L} - 1)||\nabla f(x^k) - \nabla f(w^k)|| \cdot ||\nabla f(x^k)|| - 2\tau\mathbb{E}||\nabla f(x^k)|| \cdot \sqrt{(1 - \mathbb{P}_1^k)}$$

$$(59)$$

Combining the results of the two terms, we obtain:

$$\mathbb{E}[\langle \nabla f(x^k), g^k \rangle] \geq \mathbb{E}||\nabla f(x^k)||^2 + \mathbb{E}(1 - \mathbb{P}_1^k)(\frac{C_1}{\mu} - 1)||\nabla f(x^k)|| \cdot ||\nabla f(x^k) - \nabla f(w^k)||$$

$$+ \mathbb{E}(1 - \mathbb{P}_2^k)(C_2 - 1)||\nabla f(x^k)|| \cdot ||\nabla f(w^k)|| - 2\mathbb{E}||\nabla f(x^k)|| \cdot \sqrt{(1 - \mathbb{P}_1^k)} \cdot \tau$$

$$- \mathbb{E}||\nabla f(x^k)|| \cdot \sqrt{(1 - \mathbb{P}_2^k)} \cdot \tau - \eta \cdot LG\tilde{G} \tag{60}$$

Similar to the treatment in (42) and (43), denoting $\mathbb{P}^k := \Pr(||\nabla f(x^k)|| < 3\tau \mid x^{k-1})$, we have:

$$\mathbb{E}\langle \nabla f(x^k), g^k \rangle + LG\tilde{G}\eta \geq \mathbb{E}\underbrace{(1 - (1 - \mathbb{P}^k)\frac{2\sqrt{1 - \mathbb{P}_1^k} + \sqrt{1 - \mathbb{P}_2^k}}{3})}_{0 \leq \lambda_1^k \leq 1}||\nabla f(x^k)||^2$$

$$+ \mathbb{E}\underbrace{(1 - \mathbb{P}_2^k)(C_2 - 1)}_{\lambda_2^k \geq 0}||\nabla f(x^k)|| \cdot ||\nabla f(w^k)||$$

$$+ \mathbb{E}\underbrace{(1 - \mathbb{P}_1^k)(\frac{C_1}{\mu} - 1)}_{\lambda_3^k \geq 0}||\nabla f(x^k)|| \cdot ||\nabla f(x^k) - \nabla f(w^k)|| \tag{61}$$

$$- \mathbb{E}\underbrace{\frac{\mathbb{P}^k \cdot (2\sqrt{1 - \mathbb{P}_1^k} + \sqrt{1 - \mathbb{P}_2^k})}{3}}_{0 \leq \lambda_4^k \leq 1}||\nabla f(x^k)|| \cdot 3\tau$$

For notational simplicity, we further define and restate:

$$\lambda_1^k := 1 - \frac{1}{3}(1 - \mathbb{P}^k)(2\sqrt{1 - \mathbb{P}_1^k} + \sqrt{1 - \mathbb{P}_2^k}), \quad \lambda_2^k := (1 - \mathbb{P}_2^k)(C_2 - 1), \quad \lambda_3^k := (1 - \mathbb{P}_1^k)(\frac{C_1}{L} - 1)$$

$$\lambda_4^k := \frac{1}{3}\mathbb{P}^k(2\sqrt{1 - \mathbb{P}_1^k} + \sqrt{1 - \mathbb{P}_2^k}), \quad \mathbb{P}^k := \Pr\left(||\nabla f(x^k)|| \leq 3\tau \mid x^{k-1}\right),$$

$$\mathbb{P}_1^k := \mathbb{E}_k\left[1_{\{||\nabla f_i(x^k) - \nabla f_i(w^k)|| \leq C_{1k}\}}\right], \quad \mathbb{P}_2^k := \mathbb{E}_k\left[1_{\{||\nabla f_i(w^k)|| \leq C_{2k}\}}\right].$$

Similarly, substituting into (29) and summing over the iterations, and setting $\eta = \sqrt{\frac{2(f(x^0) - f(x^*))}{TL\tilde{G}(1 + 4G)}}$, we obtain:

$$\frac{1}{T}\sum_{k=1}^{T}\mathbb{E}\left[\lambda_1^k||\nabla f(x^k)||^2 + \lambda_2^k||\nabla f(x^k)|| \cdot ||\nabla f(w^k)|| + \lambda_3^k||\nabla f(x^k)|| \cdot ||\nabla f(x^k) - \nabla f(w^k)||\right]$$

$$\leq \frac{f(x^0) - f(x^*)}{\eta T} + \frac{\eta L}{2T}\sum_{k=1}^{T}\tilde{G} + \frac{2\eta}{T}\sum_{k=1}^{T}LG\tilde{G} + \frac{1}{T}\sum_{k=1}^{T}\mathbb{E}\left[\lambda_4^k \cdot 3\tau||\nabla f(x^k)||\right]$$

$$\leq 2\sqrt{\frac{(f(x^0) - f(x^*))L\tilde{G}(1 + 4G)}{2T}} + \frac{1}{T}\sum_{k=1}^{T}\mathbb{E}\left[\lambda_4^k \cdot 3\tau||\nabla f(x^k)||\right] \tag{62}$$

## D  PROOFS OF PRIVACY ANALYSIS

In this section, we present the detailed proofs of the privacy results, i.e., Thm.3 . It is worth noting that we only discuss the privacy guarantees of DP-C4. For DP-C4$^+$, the privacy analysis is almost identical, since they share similar iterative formats. The only difference lies in the clipping coefficients $C_{1k}$ and $C_{2k}$, which leads to nearly the same conclusions. Therefore, we only present the privacy analysis for DP-C4.

## D.1 PROOF OF THEOREM 3

We utilize Rényi Differential Privacy (RDP) as a bridge to analyze the privacy guarantees of DP-C4. Our insight is that each update of DP-C4 consists of two components, namely the *Coupled Term* and the *Anchor Term*, and we allocate different privacy budget weights to these components to discuss the corresponding noise levels. We first introduce several definitions and lemmas:

**Definition 3** (Rényi Differential Privacy (RDP) (Mironov, 2017)). *A randomized mechanism $\mathcal{M}$ : $\mathcal{D} \to \mathcal{R}$ satisfies $(\alpha, \varepsilon)$-RDP ($\alpha \in (1, \infty)$, $\varepsilon > 0$) if for any datasets $D, D' \in \mathcal{D}$ with $d_{\mathrm{H}}(D, D') = 1$, it holds that*

$$\frac{1}{\alpha - 1} \log \mathbb{E}_{o \sim \mathcal{M}(D')} \left[ \left( \frac{\mathcal{M}(D)(o)}{\mathcal{M}(D')(o)} \right)^{\alpha} \right] \leq \varepsilon,$$

*where $\mathcal{M}(D)(o)$ denotes the density of $\mathcal{M}(D)$ at $o$.*

**Lemma 2** (Post-processing Property of RDP (Mironov, 2017)). *Let $\mathcal{M} : \mathcal{D} \to \mathcal{R}$ be $(\alpha, \varepsilon)$-RDP and $g : \mathcal{R} \to \mathcal{R}'$ be any function. Then the composed mechanism $g \circ \mathcal{M} : \mathcal{D} \to \mathcal{R}'$ is also $(\alpha, \varepsilon)$-RDP.*

**Lemma 3** (Composition of RDP Mechanisms (Mironov, 2017)). *Let $\mathcal{M}_r : \mathcal{R}_1 \times \cdots \times \mathcal{R}_{r-1} \times \mathcal{D} \to \mathcal{R}_r$ be $(\alpha, \varepsilon_r)$-RDP for $r \in [R]$. Then the mechanism*

$$\mathcal{M}(D) := (\mathcal{M}_1(D), \mathcal{M}_2(\mathcal{M}_1(D), D), \ldots, \mathcal{M}_R(\mathcal{M}_1(D), \ldots, D))$$

*is $(\alpha, \sum_{r=1}^{R} \varepsilon_r)$-RDP.*

**Lemma 4** (Conversion from RDP to DP (Mironov, 2017)). *If a mechanism $\mathcal{M}$ is $(\alpha, \varepsilon)$-RDP, then $\mathcal{M}$ also satisfies $(\varepsilon + \frac{\log(1/\delta)}{\alpha - 1}, \delta)$-DP for any $\delta \in (0, 1)$.*

**Lemma 5** (Gaussian Mechanism (Mironov, 2017)). *Given a function $h$, the Gaussian Mechanism*

$$\mathcal{M}(D) := h(D) + \mathcal{N}(0, \sigma^2 I)$$

*satisfies $(\alpha, \alpha \Delta^2(h)/(2\sigma^2))$-RDP for every $\alpha \in (1, \infty)$.*

With these preparations, we first analyze the sensitivity of each component in DP-C4$^{(+)}$. We have the following lemma:

**Lemma 6** ($\ell_2$-sensitivity). *In Algorithm 3, the sensitivities of the Coupled Term $g_1^k$ and the Anchor Term $g_2^k$ are given by*

$$\Delta_{1k} = \frac{2C_{1k}}{|S|}, \qquad \Delta_{2k} = \frac{2C_{2k}}{|D|}.$$

*Proof.* For the Coupled Term, we have:

$$g_1^k = \frac{1}{|S|} \sum_{i \in S} \mathrm{clip}\left( \nabla f_i(x^k) - \nabla f_i(w^k), \, C_{1k} \right).$$

The $\ell_2$-sensitivity of $g_1^k$ is bounded by

$$\max_{S, S'} \| g_1^k - g_1'^k \| = \max_{S, S'} \left\| \frac{1}{|S|} \sum_{i \in S} clip(\nabla f_i(x^k) - \nabla f_i(w^k)) - \frac{1}{|S'|} \sum_{i \in S'} clip(\nabla f_i(x^k) - \nabla f_i(w^k)) \right\|$$

$$= \max_{S, S'} \frac{1}{|S|} \| clip(\nabla f_j(x^k) - \nabla f_j(w^k)) - clip(\nabla f_j'(x^k) - \nabla f_j'(w^k)) \|$$

$$= \max_{S, S'} \frac{1}{|S|} \left\| \min\left\{ \frac{C_{1k}}{\|\nabla f_j(x^k) - \nabla f_j(w^k)\|}, 1 \right\} (\nabla f_j(x^k) - \nabla f_j(w^k)) \right.$$

$$\left. - \min\left\{ \frac{C_{1k}}{\|\nabla f_j'(x^k) - \nabla f_j'(w^k)\|}, 1 \right\} (\nabla f_j'(x^k) - \nabla f_j'(w^k)) \right\|$$

$$\leq \max_{S, S'} \frac{1}{|S|} \left( \left\| \min\left\{ \frac{C_{1k}}{\|\nabla f_j(x^k) - \nabla f_j(w^k)\|}, 1 \right\} (\nabla f_j(x^k) - \nabla f_j(w^k)) \right\| \right.$$

$$\left. + \left\| \min\left\{ \frac{C_{1k}}{\|\nabla f_j'(x^k) - \nabla f_j'(w^k)\|}, 1 \right\} (\nabla f_j'(x^k) - \nabla f_j'(w^k)) \right\| \right)$$

$$\leq \frac{2C_{1k}}{|S|} := \Delta_{1k}.$$

For the Anchor Term, we have:

$$g_2^k = \frac{1}{|D|} \sum_{i \in D} clip(\nabla f_i(w^k), C_{2k}).$$

The $\ell_2$-sensitivity of $g_2^k$ can be bounded as

$$\max_{D,D'} \|g_2^k - g_2'^k\| = \max_{D,D'} \left\| \frac{1}{|D|} \sum_{i \in D} \nabla f_i(w^k) - \frac{1}{|D'|} \sum_{i \in D'} \nabla f_i(w^k) \right\|$$

$$= \max_{D,D'} \frac{1}{|D|} \|\nabla f_j(w^k) - \nabla f_j'(w^k)\|$$

$$= \max_{D,D'} \frac{1}{|D|} \left\| \min\left\{ \frac{C_{2k}}{\|\nabla f_j(w^k)\|}, 1 \right\} \nabla f_j(w^k) - \min\left\{ \frac{C_{2k}}{\|\nabla f_j'(w^k)\|}, 1 \right\} \nabla f_j'(w^k) \right\|$$

$$\leq \max_{D,D'} \frac{1}{|D|} \left( \left\| \min\left\{ \frac{C_{2k}}{\|\nabla f_j(w^k)\|}, 1 \right\} \nabla f_j(w^k) \right\| + \left\| \min\left\{ \frac{C_{2k}}{\|\nabla f_j'(w^k)\|}, 1 \right\} \nabla f_j'(w^k) \right\| \right)$$

$$\leq \frac{2C_{2k}}{|D|} := \Delta_{2k}.$$

$\square$

With all the necessary preparations in place, we now proceed to the next step. We focus on analyzing Routines 1 and 2; the analysis for the remaining paths is similar, yielding the same conclusions. First, we derive an RDP bound for each term $g_1^k$ and $g_2^k$.

For $g_1^k$, from Lemma.5, when we add noise $\mathbf{n}_1^k \sim \mathcal{N}(0, \sigma_1^2 C_{1k}^2)$, the term $g_1^k$ satisfies $(\alpha, 2\alpha/(\sigma_1^2 \cdot |S|^2))$-RDP, where the sensitivity of $g_1^k$ is given in Lemma.6.

Similarly, for $g_2^k$, from Lemma.5, when we add noise $\mathbf{n}_2^k \sim \mathcal{N}(0, \sigma_2^2 C_{2k}^2)$, the term $g_2^k$ satisfies $(\alpha, 2\alpha/(\sigma_2^2 \cdot |D|^2))$-RDP, where the sensitivity of $g_2^k$ is given in Lemma.6.

From Lemma.3, Alg.3 satisfies

$$(\alpha, \frac{2\alpha T}{\sigma_1^2 \cdot |S|^2} + \frac{2\alpha T p}{\sigma_2^2 \cdot |D|^2})\text{-RDP}.$$

Then, by Lemma.4, it follows that Algorithm 3 satisfies

$$\left( \frac{2\alpha T}{\sigma_1^2 |S|^2} + \frac{2\alpha T p}{\sigma_2^2 |D|^2} + \frac{\log(1/\delta)}{\alpha - 1}, \delta \right)\text{-DP}.$$

For any target DP parameters $(\epsilon_{DP}, \delta_{DP})$, we discuss the variance of these noises through the allocation of the privacy budget. We set:

$$\begin{cases} \frac{1}{2}\epsilon_{DP} = \frac{\log(1/\delta)}{\alpha - 1} \\ \frac{1}{2}\epsilon_{DP} = \frac{2\alpha T}{\sigma_1^2 |S|^2} + \frac{2\alpha T p}{\sigma_2^2 |D|^2} \\ \delta_{DP} = \delta \end{cases} \tag{63}$$

From the first line of the above equation, we obtain $\alpha = 1 + 2\log(1/\delta_{DP})/\epsilon_{DP}$. In the following, under the constraint $\frac{1}{2}\epsilon_{DP} = \frac{2\alpha T}{\sigma_1^2 |S|^2} + \frac{2\alpha T p}{\sigma_2^2 |D|^2}$, we aim to minimize the total noise magnitude added to the gradient estimator per iteration, i.e., $\sigma_1^2 + \sigma_2^2$.

Let $\frac{2\alpha T}{\sigma_1^2 |S|^2} = \frac{1}{2}\beta\epsilon_{DP}$, $\frac{2\alpha T p}{\sigma_2^2 |D|^2} = \frac{1}{2}(1-\beta)\epsilon_{DP}$, where $\beta \in (0, 1)$. Solving for $\sigma_1^2$ and $\sigma_2^2$ yields:

$$\sigma_1^2 = \frac{4\alpha T}{\beta |S|^2 \epsilon_{DP}}, \quad \sigma_2^2 = \frac{4\alpha T p}{(1 - \beta)|D|^2 \epsilon_{DP}} \tag{64}$$

Continuing the above objective, we aim to minimize the total noise per step by adjusting the budget allocation coefficient $\beta$, i.e., $\min_\beta \sigma_1^2 + \sigma_2^2$, and let $\theta = \frac{|D|^2}{|S|^2} \geq 1$. That is,

$$\min_{\beta \in (0,1)} \frac{1}{\beta} + \frac{p}{(1-\beta)\theta} := y \tag{65}$$

Taking the derivative with respect to $\beta$ and setting it to zero, we obtain:

$$\frac{dy}{d\beta} = -\frac{1}{\beta^2} + \frac{p}{\theta(1-\beta)^2} = 0 \tag{66}$$

Solving this, we obtain the value of $\beta$ that minimizes $\min_\beta \sigma_1^2 + \sigma_2^2$ as:

$$\beta^* = \frac{1}{1 + \sqrt{\frac{p}{\theta}}} \tag{67}$$

Substituting back into (64), we obtain:

$$\sigma_1^2 = \frac{4T(2\log(1/\delta_{DP}) + \epsilon_{DP})}{|S|^2 \epsilon_{DP}^2} \cdot (1 + \sqrt{\frac{p}{\theta}}),$$

$$\sigma_2^2 = \frac{4T(2\log(1/\delta_{DP}) + \epsilon_{DP})}{|D|^2 \epsilon_{DP}^2} \cdot \sqrt{p} \cdot (\sqrt{\theta} + \sqrt{p}) \tag{68}$$

$$= \frac{4T(2\log(1/\delta_{DP}) + \epsilon_{DP})}{|S|^2 \epsilon_{DP}^2} \cdot (\frac{p}{\theta} + \sqrt{\frac{p}{\theta}})$$

Let $\sigma^2 = \frac{4T(2\log(1/\delta_{DP}) + \epsilon_{DP})}{|S|^2 \epsilon_{DP}^2}$. It is straightforward to see that $\sigma^2 = \sigma_{\text{DP-SGD}}^2$ coincides exactly with the noise magnitude used in DP-SGD. In summary, we have:

$$(\sigma_1^2, \sigma_2^2)_{Routine\ 1\&2} = \left((1 + \sqrt{\frac{p}{\theta}})\sigma^2, (\frac{p}{\theta} + \sqrt{\frac{p}{\theta}})\sigma^2\right) \tag{69}$$

For $(\sigma_1^2 + \sigma_2^2)_{Routine\ 1\&2}$, since $\frac{p}{\theta}$ is very small, we have:

$$(\sigma_1^2 + \sigma_2^2)_{Routine\ 1\&2} = \left(1 + \sqrt{\frac{p}{\theta}} + \frac{p}{\theta} + \sqrt{\frac{p}{\theta}}\right)\sigma^2$$
$$= (1 + \sqrt{\frac{p}{\theta}})^2 \cdot \sigma^2 \approx \sigma^2 \tag{70}$$

For Routines 3 and 4, since $g_1^k = 0$ when $w^{k+1} = x^{k+1}$, under $T$ iterations, we only compute $g_1^k$ for $T(1-p)$ rounds. Similarly, we can obtain:

$$(\sigma_1^2, \sigma_2^2)_{Routine\ 3\&4} = \left((1-p+\sqrt{\frac{p(1-p)}{\theta}})\sigma^2, (\frac{p}{\theta}+\sqrt{\frac{p(1-p)}{\theta}})\sigma^2\right) \tag{71}$$

For $(\sigma_1^2 + \sigma_2^2)_{Routine\ 3\&4}$, let $p = \frac{2|S|}{|D|}$, we have:

$$(\sigma_1^2 + \sigma_2^2)_{Routine\ 3\&4} = (1-p+\sqrt{\frac{p(1-p)}{\theta}} + \frac{p}{\theta} + \sqrt{\frac{p(1-p)}{\theta}})\sigma^2$$
$$= (\sqrt{1-p} + \sqrt{\frac{p}{\theta}})^2 \cdot \sigma^2$$
$$= (1 - \frac{p}{2} - \frac{p^2}{8} - O(p^3) + \frac{p^{\frac{3}{2}}}{2}) \cdot \sigma^2 \tag{72}$$
$$= (1 - (\frac{p}{2} + \frac{p^2}{8}) - O(p^3) + \frac{p^{\frac{3}{2}}}{2}) \cdot \sigma^2$$
$$\leq (1 - O(p^3)) \cdot \sigma^2 < \sigma^2$$

For comparison, in the case of DP-SVRG, The noise added to the gradient estimator consists of three components. Similarly, we can derive that:

$$(\sigma_1^2 + \sigma_2^2 + \sigma_3^2)_{Routine\ 1\&2}^{DP-SVRG} = (1 + \sqrt{\frac{2p}{\theta}})^2 \cdot \sigma^2 > \sigma^2$$

$$(\sigma_1^2 + \sigma_2^2 + \sigma_3^2)_{Routine\ 3\&4}^{DP-SVRG} = (\sqrt{2(1-p)} + \sqrt{\frac{p}{\theta}})^2 \cdot \sigma^2 > \sigma^2 \tag{73}$$

From this, the multipliers for each kind of noise are summarized in the following table:

Table 3: Routine 1&2

| Methods | $\sigma^2_{DP-SGD}$ | $\sigma^2_{DP-SVRG}$ | $\sigma^2_{DP-C4(+)}$ |
|---|---|---|---|
| Noise Multiplier | $\sigma^2$ | $\sigma^2 \cdot (1+\sqrt{\frac{2p}{\theta}})^2$ | $\sigma^2 \cdot (1+\sqrt{\frac{p}{\theta}})^2$ |
| Comparison | $\sigma^2_{DP-SGD} = \sigma^2$ | $\sigma^2_{DP-SVRG} > \sigma^2$ | $\sigma^2_{DP-C4(+)} \approx \sigma^2$ |

Table 4: Routine 3&4

| Methods | $\sigma^2_{DP-SGD}$ | $\sigma^2_{DP-SVRG}$ | $\sigma^2_{DP-C4(+)}$ |
|---|---|---|---|
| Noise Multiplier | $\sigma^2$ | $\sigma^2 \cdot (\sqrt{2(1-p)}+\sqrt{\frac{p}{\theta}})^2$ | $\sigma^2 \cdot (\sqrt{1-p}+\sqrt{\frac{p}{\theta}})^2$ |
| Comparison | $\sigma^2_{DP-SGD} = \sigma^2$ | $\sigma^2_{DP-SVRG} > \sigma^2$ | $\sigma^2_{DP-C4(+)} < \sigma^2$ |

# E    ADDITIONAL EXPERIMENTS

In this section, we provide additional information and results on our numerical experiments that are not given in the main paper due to the space limitation.

**Datasets Information**    We conduct experiments on Mushroom, MNIST, CIFAR-10, CIFAR-100, IMDb, and GLUE-SST-2. The information of all datasets used is summarized in Table 5.

Table 5: The summary of the datasets used in the experiments.

| Dataset | Samples | Type | Classes | Task |
|---|---|---|---|---|
| Mushroom | 8,124 | Tabular | 2 | SVM |
| MNIST | 70,000 | Image (28×28, Gray) | 10 | CV |
| CIFAR-10 | 60,000 | Image (32×32, RGB) | 10 | CV |
| CIFAR-100 | 60,000 | Image (32×32, RGB) | 100 | CV |
| IMDb | 50,000 | Text (Reviews) | 2 | NLP |
| GLUE-SST-2 | 67,349 | Text (Sentences) | 2 | NLP |

**Results on Different $C_1$ and $C_2$**    First, we provide an ablation study on the selection of clipping thresholds $C_1$ and $C_2$. We conduct experiments on the CIFAR-10 dataset with the learning rate set to $\eta = 0.025$ and the privacy parameter $(\epsilon, \delta) = (5, 10^{-5})$. Following the main experiment, we set the mini-batch size to $|S| = 256$, the large-batch size to $|D'| = 4096$ and $p = \frac{2|S|}{|D'|} = 0.125$. We fix $C = 1$ and vary $C_1$ and $C_2$ over the range $\{0.125, 0.25, 0.5, 1, 2, 4, 8, 16, 32, 64\}$. We report the results for each configuration and compare them against DP-SGD. The experimental results of DP-C4 and DP-C4$^+$ are presented in Figure.2 and Figure.3, respectively. In each cell of the heatmap, the color encodes the corresponding accuracy, with warmer shades indicating higher accuracy and cooler shades indicating lower accuracy. Each cell further reports the accuracy associated with the corresponding clipping thresholds, while the value in parentheses denotes the accuracy difference relative to DP-SGD.

On the one hand, for both DP-C4 and DP-C4$^+$, when examining a single row or column of the grid, we observe that increasing $C_1$ initially improves accuracy, which subsequently decreases; a similar trend is observed when increasing $C_2$. More specifically, as $C_1$ and $C_2$ gradually increase, the injected noise becomes larger, leading to a gradual degradation in accuracy until it converges to a constant value. In particular, when $C_1 = C_2 = 64$, the accuracies of both DP-C4 and DP-C4$^+$ converge to 61.16, since in this case a constant clipping threshold is applied at each iteration (i.e., in DP-C4: $C_{1k} = \min\{C, C_1\|\Delta_S^k\|\} = C$, $C_{2k} = \min\{C, C_2\|\nabla f(w^k)\|\} = C$; in DP-C4$^+$: $C_{1k} = \min\{C, C_1\|x^k - w^k\|\} = C$, $C_{2k} = \min\{C, C_2\|\nabla f(w^{k-1})\|\} = C$).

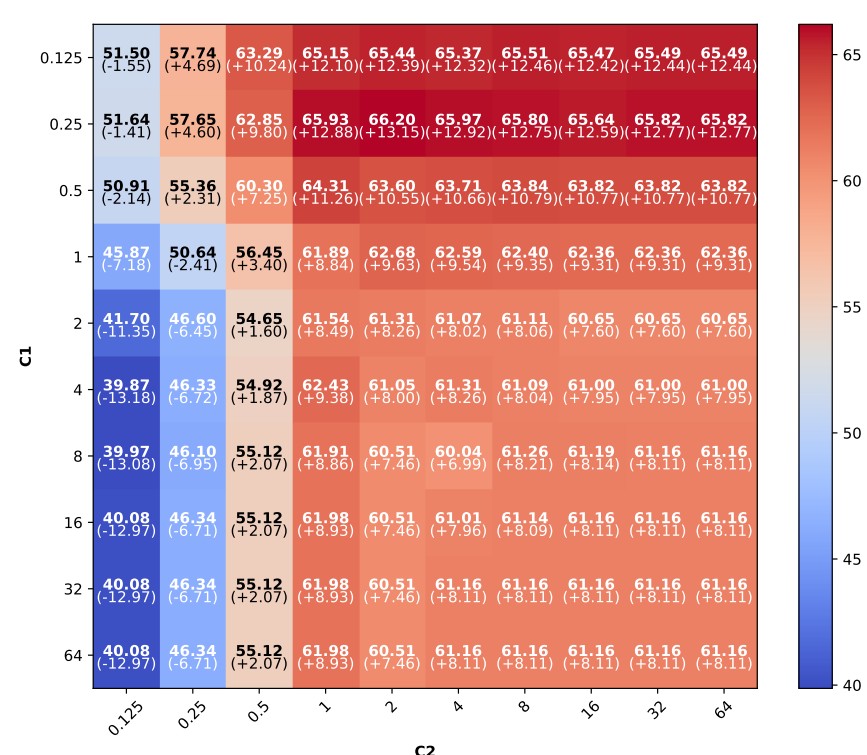

Figure 2: Accuracy of DP-C4 with different $C_1$ and $C_2$

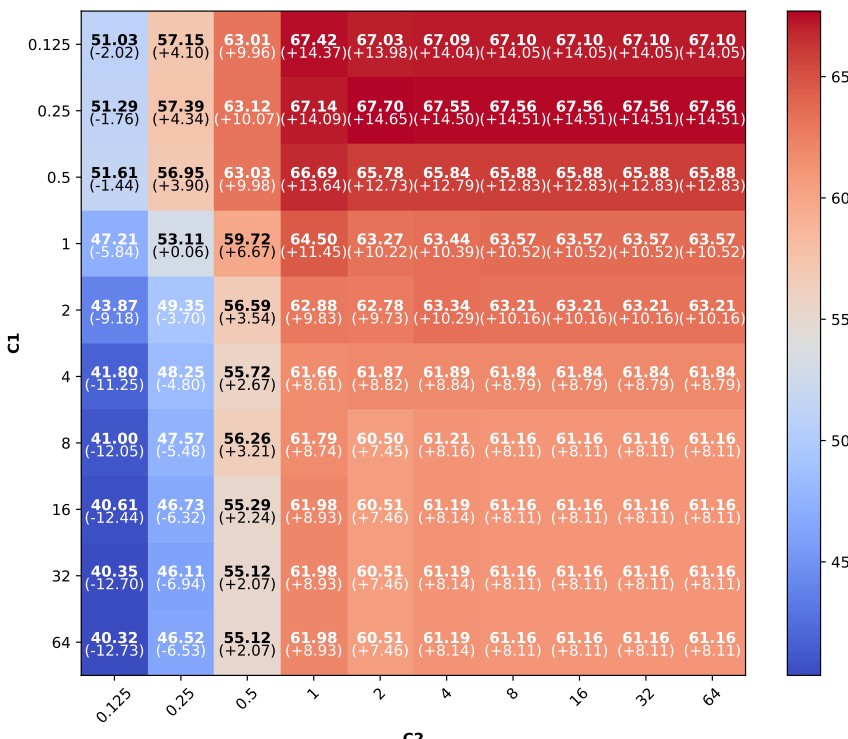

Figure 3: Accuracy of DP-C4$^+$ with different $C_1$ and $C_2$

On the other hand, when $C_1$ is small, the accuracy does not decrease significantly. This is because as $C_1 \rightarrow 0$, the coupled term of DP-C4$^{(+)}$ vanishes, which essentially reduces the method to a large-batch variant of delayed DP-SGD. The iterative structure is thus not severely disrupted, while the injected noise is substantially reduced. In contrast, when $C_2$ is small, accuracy drops sharply. This is due to the excessive clipping bias, which prevents effective updates (i.e., $\frac{1}{|S|} \sum_{i \in S} \text{clip}(\nabla f_i(x^k) - \nabla f_i(w^k)) + \frac{1}{|D|} \sum_{i \in D} \text{clip}(\nabla f_i(w^k)) \approx \frac{1}{|S|} \sum_{i \in S} \text{clip}(\nabla f_i(x^k) - \nabla f_i(w^k)))$. In summary, the vanishing of the coupled term can be tolerated since it still preserves an effective optimization structure, whereas the vanishing of the anchor term is detrimental, as it leads to severe performance degradation.

**Results on Different $C$**  We also conduct an ablation study on the overall clipping threshold $C$. The experiments are performed on CIFAR-10 with $\eta = 0.025$, $|S| = 256$, $|D'| = 4096$, the privacy parameter $(\epsilon, \delta) = (5, 10^{-5})$, and $p = 0.125$. We fix $C_1 = C_2 = 1$ and vary $C$ over the set $\{0.125, 0.25, 0.5, 1, 2, 4, 8, 16, 32, 64\}$. The results comparing DP-C4$^{(+)}$ with DP-SGD are summarized in Table 6.

Table 6: Test accuracy of different methods on different clipping threshold $C$.

| Method | Values of Clipping Threshold $C$ | | | | | | | | | |
|---|---|---|---|---|---|---|---|---|---|---|
| | 0.125 | 0.25 | 0.5 | 1 | 2 | 4 | 8 | 16 | 32 | 64 |
| DP-SGD | 54.91 | 55.36 | 53.42 | 53.05 | 41.01 | 27.40 | 18.36 | 16.41 | 14.93 | 10.74 |
| DP-C4 | 55.39 | 59.91 | 62.78 | 61.89 | 59.84 | 59.80 | 59.65 | 59.65 | 59.65 | 59.65 |
| DP-C4$^+$ | 55.84 | 59.22 | 61.52 | 64.50 | 61.30 | 52.81 | 42.14 | 34.89 | 28.25 | 24.41 |

It can be observed that, on the one hand, as $C$ decreases, the accuracy of both DP-SGD and DP-C4$^{(+)}$ first increases and then decreases. This behavior is attributed to the reduction of the injected noise and the simultaneous growth of the clipping bias. When $C$ becomes sufficiently small, every term is clipped on a per-sample basis, and thus the iterations of all three methods resemble a normalized update scheme. On the other hand, as $C$ increases, the accuracy of DP-SGD drops rapidly, while that of DP-C4$^+$ decreases more slowly, and DP-C4 eventually converges to a fixed accuracy level of 59.65%. This robustness stems from the fact that the effective clipping thresholds of DP-C4 are determined by $C_{1k} = \min\{C, C_1 \|\nabla f_S(x^k) - \nabla f_S(w^k)\|\} \leq 2C_1 G$, $C_{2k} = \min\{C, C_2 \|\nabla f(w^k)\|\} \leq C_2 G$, which are governed by the gradient difference and the full gradient, and therefore do not grow unbounded. In contrast, for DP-C4$^+$, the clipping coefficient of the coupled term is given by $C_{1k} = \min\{C, C_1 \|x^k - w^k\|\}$, as the iterations proceed, $\|x^k - w^k\|$ may occasionally become relatively large with non-negligible probability, which in turn introduces a larger amount of noise and leads to performance degradation.

**Results on Different Routine**  We further conduct experiments on CIFAR-10 using different routines. We fix $C = C_1 = C_2 = 1$, while keeping the remaining parameters unchanged. The results are reported in Table 7.

Table 7: Test accuracy of DP-C4$^{(+)}$ on different routines.

| Method | Different Routines | | | |
|---|---|---|---|---|
| | 1 | 2 | 3 | 4 |
| DP-C4 | 61.89 | 62.16 | 61.23 | 62.10 |
| DP-C4$^+$ | 64.50 | 64.39 | 63.97 | 64.28 |

We observe that the results of the four routines are similar. This is because the different routines only modify the update strategy of $w^k$ and do not alter the intrinsic properties of the DP-C4$^{(+)}$ iterative scheme, so that their behavior is largely similar in expectation.

**Results on Different Large Batchsizes**  We conduct experiments on CIFAR-10 using DP-C4$^{(+)}$ under different large-batch sizes. The learning rate is set to $\eta = 0.025$, with $C = C_1 = C_2 = 1$,

$|S| = 256$, and $p = 0.125$. We vary the large-batch size as $|D'| \in \{512 = 2 \cdot |S|, 2^2 \cdot |S|, 2^3 \cdot |S|, 2^4 \cdot |S|, 2^5 \cdot |S| = 8192\}$, and record the corresponding accuracies of DP-C4$^{(+)}$. The detailed results are presented in Table 8.

Table 8: Test accuracy of DP-C4$^{(+)}$ on different large-batch sizes.

| Method | Different Large-batch Sizes | | | | |
|---|---|---|---|---|---|
| | 512 | 1024 | 2048 | 4096 | 8192 |
| DP-C4 | 44.18 | 52.60 | 58.68 | 61.89 | 59.93 |
| DP-C4$^+$ | 41.12 | 52.99 | 60.98 | 64.50 | 58.91 |

We observe that as $|D'|$ increases, the accuracy of DP-C4$^{(+)}$ first rises and then decreases. This behavior occurs because a relatively small large batch leads to inaccurate estimation of the full gradient and, compared to DP-SGD, introduces excessive clipping bias. Conversely, an excessively large batch significantly increases the number of samples averaged in each iteration, which effectively reduces the number of updates and consequently degrades performance.

**Results on Different** $p$ We conducted experiments on CIFAR-10 to evaluate DP-C4$^{(+)}$ under different update probabilities $p$. We set the learning rate to $\eta = 0.025$, with $C = C_1 = C_2 = 1$, $|S| = 256$, and $|D'| = 4096$. We varied $p \in \{\frac{1}{2}, \frac{1}{2^2}, \frac{1}{2^3}, \frac{1}{2^4}, \frac{1}{2^5}\}$ and recorded the corresponding accuracy of DP-C4$^{(+)}$. The detailed results are presented in Table 9.

Table 9: Test accuracy of DP-C4$^{(+)}$ on different $p$.

| Method | Different $p$ | | | | |
|---|---|---|---|---|---|
| | 0.5 | 0.25 | 0.125 | 0.0625 | 0.03125 |
| DP-C4 | 58.92 | 60.83 | 61.89 | 63.40 | 61.76 |
| DP-C4$^+$ | 60.04 | 63.67 | 64.50 | 63.65 | 62.71 |

We can observe that as $p$ decreases, the accuracy of DP-C4$^{(+)}$ first increases and then decreases. This phenomenon can be explained as follows: when $p$ is relatively large, the anchor term is updated frequently, which increases the average data consumption per iteration and consequently reduces the effective number of iterations, leading to suboptimal performance. On the other hand, when $p$ is too small, the anchor term is updated too infrequently, which also negatively impacts the accuracy.

