# OpenReview forum: "DP-C4: Eliminating Solution Bias in Differentially Private Optimization via Coupled Clipping with Adaptive Thresholds"
_ICLR.cc/2026/Conference — ICLR 2026 Conference Withdrawn Submission_

### Official Review · Reviewer_Y9J6 · 2025-10-31

**Soundness:** 3
**Presentation:** 3
**Contribution:** 2
**Rating:** 4
**Confidence:** 3

**Summary:**

This paper proposes DP-C4 framework that employs coupled clipping of gradient differences with adaptive thresholds. The key innovation is ensuring both clipping bias and noise variance asymptotically vanish, attempting convergence in strongly convex settings and to a diminishing neighborhood in nonconvex cases. The authors provide supportive theoretical convergence guarantees, privacy analysis and empirical validation across several tasks.

**Strengths:**

1. It tackles the limitation of DP optimization the inherent solution bias caused by constant clipping and noise. This is a practically relevant problem that has received limited attention in prior works.
2. The fixed-point analysis in Section 3.3 demonstrates why DP-C4 eliminates solution bias while existing methods do not. Also, the Lyapunov functions (Theorem-1) for analyzing DP algorithms is promising.
3. It includes ablation studies examining the impact of key hyperparameters including the overall clipping threshold $C$, different anchor update routines (R1- R4), large batch sizes $|D'|$ and update probability $p$. These analysis provides insights into the robustness and sensitivity of  DP-C4 and  DP-C4+ across parameter choices.

**Weaknesses:**

1. The core idea closely follows DP-SVRG by clipping gradient differences $\nabla f_i(x^k) - \nabla f_i(w^k)$. While the coupled clipping with adaptive thresholds is new, the fundamental structure is borrowed from variance reduction techniques. The paper should more clearly articulate what is genuinely novel beyond adaptive threshold selection.
2. A thorough comparison table is missing, showing Big-O complexity bounds (iteration complexity, sample complexity, etc.) alongside recent SOTA methods (e.g., DP-Adam, DP-SCAFFOLD) and other variance-reduced DP methods. Without such a table, it is impossible to justify the theoretical improvements and the additional computational overhead. The paper claims memory efficiency for DP-C4$^+$, however, it lacks formal complexity analysis or empirical runtime comparisons.
3. The convergence rates in Theorems 1-2 are not explicitly stated in standard $\mathcal{O}(\cdot)$ notation and compared against known rates for DP-SGD ($\mathcal{O}(1/\sqrt{T})$ for nonconvex and DP-SVRG. Therefore, it is unclear whether the asymptotic solution bias elimination comes at the cost of slower convergence or higher per-iteration complexity.
4. The results section is not sufficiently strong to support the claims. Table-1 compares methods at fixed $C=1$, but doesn't control for the actual privacy budget ($\varepsilon$ ) consumed. Different methods may use different $\varepsilon$ to achieve these results, making the comparison unfair.
5. No analysis of how performance varies with privacy budget ($\varepsilon$). Also, there is limited discussion of how DP-C4 relates to recent adaptive methods beyond DiceSGD.

**Questions:**

1. Could you clarify how to implement the gradual increase of $C_2$ while maintaining the convergence guarantee from Theorem-1? Given that the theorem requires $C_2 \geq \frac{\tau}{e}+1$ and $\min { ||\nabla f(w^k)||, ||x^k - x^*||}> e$, what algorithm or strategy would you recommend for adapting $C_2$ during training to avoid excessive noise injection in early iterations?
2. The paper claims DP-C4 achieves "the same level of privacy protection while adding less noise'' compared to DP-SGD. Could you provide experiment results with privacy-utility curves at equal ($\varepsilon$, $\delta$) budgets with optimally tuned hyperparameters for each method?
3. The choice of large batch size $|D'| = 4096$ is not justified clearly. How sensitive are results to batch size?
4. The nonconvex convergence result in Equation (7) involves separate $\lambda$ coefficients ($\lambda_1^k$, $\lambda_2^k$, $\lambda_3^k$, $\lambda_4^k$) with complex dependencies on $\mathbb{P}^k$, $\mathbb{P}_1^k$, $\mathbb{P}_2^k$, which is difficult to interpret. Why are so many separate terms necessary? Could these $\lambda$ coefficients be combined into fewer or more interpretable quantities? The claim that bias "gradually vanishes'' appears to rely on taking $C_2 \to \infty$, it directly contradicts the noise minimization objective since larger clipping thresholds require more noise for privacy. How to set $C_1$ and $C_2$ to balance these competing objectives?
5. The fixed-point analysis in Section 3.3 assumes convergence has occurred, but doesn't address the transient behavior. Does the fixed-point analysis provide any rates of convergence or characterization of the transient behavior before reaching the fixed point?

---

### Official Review · Reviewer_Rr8d · 2025-10-31

**Soundness:** 1
**Presentation:** 2
**Contribution:** 1
**Rating:** 0
**Confidence:** 4

**Summary:**

This manuscript addresses the problem of mitigating both clipping bias and noise variance to eliminate solution bias in differentially private (DP) stochastic optimization algorithms. The authors propose a novel framework, DP-C4, to tackle the solution bias introduced by per-sample gradient clipping and noise injection used for protecting sensitive information. DP-C4 employs coupled per-sample gradient clipping with adaptively chosen thresholds to reduce such bias. The authors prove that the proposed method can simultaneously ensure the asymptotic vanishing of clipping bias and noise variance without compromising privacy protection, achieving the same privacy level as DP-SGD. They also provide a novel convergence analysis for both the strongly convex case (proved to converge to the optimum) and the nonconvex case (shown to converge to a diminishing neighborhood of a first-order stationary point). Moreover, to reduce the memory burden of DP-C4, they propose DP-C4+, an extended version that removes the need to store all gradients involved in the computation while still maintaining theoretical properties of DP-C4. Finally, they present empirical validations and ablation studies demonstrating that DP-C4 and DP-C4+ achieve better privacy-utility trade-offs than existing baselines across multiple datasets and tasks.

**Strengths:**

This manuscript tries to address an emerged  yet crucial issue in DP optimization, the joint mitigation of clipping bias and noise variance, where most existing DP algorithms (e.g., DP-SGD, DP-SVRG) tend to sacrifice one for the other.

**Weaknesses:**

The privacy analysis of this paper seems problematic.

Step 11 in the main algorithm does not satisfy DP guarantees. The C_1 and C_2 have not been privately released. Also, it seems that the author claims to add O(1) noise to the gradient average instead of O(1/n) noise expected in standard DP-SGD. I am afraid that the SGD cannot tolerate such a noise which is even larger than the gradient signal.

**Questions:**

Can the author explain or justify the noise parameter selection detailed in the paper with a formal proof?

---

### Official Review · Reviewer_cv9R · 2025-10-31

**Soundness:** 1
**Presentation:** 3
**Contribution:** 2
**Rating:** 2
**Confidence:** 1

**Summary:**

The paper focus on debiasing the clipping bias in DP optimization. The paper proposes clipping the gradient, and difference of gradient at two points, and adopts an SVRG-like update to reduce the variance.

The paper provides the convergence and privacy analysis for the proposed method under standard assumptions. By constructing a Lyaunov  function, the paper shows that the proposed method converges to a neighborhood of the stationary point with clipping.

Numerical results shows that the proposed method achieves a better performance compared with existing algorithms.

**Strengths:**

1. The paper proposed a novel algorithm that converges to the neighbourhood the the statioanry point under clipping and without assumming gradient distribution.

2. The paper is clearly written with sufficient details explaining the algorithm and its theoretical properties.

**Weaknesses:**

1. The privacy analysis does not seems to be correct. The clipping threshold $C_{1k}, C_{2k}$ are computed using private data, but they are not privatized by the algorithm. More specifically, the sensitivity of the clipped gradient in the analysis provided in the appendix is incorrect, because the change of a sample not only impact the clipped gradient $g_1, g_2$, but also impact $C_{1}, C_2$. However, the privacy analysis treats the clipping threshold as non-private information and directly use them to determine the size of the noise.

**Questions:**

Please address the potential mistake in the privacy guarantee.

---

### Official Review · Reviewer_73q5 · 2025-11-07

**Soundness:** 2
**Presentation:** 2
**Contribution:** 2
**Rating:** 2
**Confidence:** 3

**Summary:**

DP-C4 and DP-C4+ are introduced for differentially private (DP) optimization. The core idea is to use a variance-reduction method for gradient estimation to enhance utility in DP algorithms. The empirical results are promising and convergence guarantees are established. Experiments are carried out on SVM, image classification, and NLP tasks.

**Strengths:**

- The empirical results are promising and convergence guarantees are established.
- Experiments are carried out on SVM, image classification, and NLP tasks.

**Weaknesses:**

- Baselines are not the latest approaches, and the superiority on the performance is limited. No results on the memory usage for DP-C4+.
- The assumptions on bounded gradients, L-smoothness, and (µ-)strong convexity mostly don't hold for deep structures.

**Questions:**

- introducing more up-to-date baselines,
- the backbone methods need to conider more modern architectures.
- can he assumptiosn on bounded gradients, L-smoothness, and (µ-)strong convexity be relaxed?

---

### Note · Authors · 2025-11-13

I have read and agree with the venue's withdrawal policy on behalf of myself and my co-authors.